

# Enhancement of biogenic emissions of VOCs in the semi-arid region of India during winter to summer transition period: Role of meteorological conditions

Nidhi Tripathi[1,2] and Lokesh Kumar Sahu[1*]

[1]Physical Research Laboratory (PRL), Navrangpura, Ahmedabad, 380009, India

[2]Indian Institute of Technology Gandhinagar Palaj, Gandhinagar, India

*Correspondence to*: L. K. Sahu (lokesh@prl.res.in)

## Abstract

Emissions of biogenic volatile organic compounds (BVOCs) play important roles in ecophysiology and atmospheric chemistry at large spatial and temporal scales. Tropical regions are a main global source of

BVOCs and magnitude and chemical compositions are highly variable. This study is based on the measurements of monoterpenes using proton transfer reaction-time of flight-mass spectrometer (PTR-TOF-MS) at a semi-arid site in western India during the winter-to-summer transition. Mixing ratio of monoterpenes showed strong diurnal variation with elevated values from evening till midnight and lowest in the afternoon. The daily data does not show clear trends with monthly means of ~0.35 ppbv

during each month. Exceptionally high levels of 3-6 ppbv were measured during the sporadic biomass burning and bonfire event during *Holi* festival. The daytime data of monoterpenes do not clearly reflect the impact of biogenic emission due to the competing influences of mixing and OH-reaction loss. In the afternoon, the monoterpenes/benzene ratio of 0.43 ppbv ppbv$^{-1}$ in second half of March was ~3 times higher than that in first half of February. It showed strong response with temperature as it increased from

0.27 ppbv ppbv$^{-1}$ (<30$^{o}$C) to 0.50 ppbv ppbv$^{-1}$ (>30$^{o}$C). The dependence with wind speed is represented by exponential decay but rate of decline in February was ~2 times greater than that in March. The ratios of monoterpenes/isoprene in the night were significant higher than those during the day indicating light independent but temperature dependent emissions of monoterpenes. The nighttime MTs/isoprene ratio increased from 0.25 ppbv ppbv$^{-1}$ in the first half of February to 0.43 ppbv ppbv$^{-1}$ in the second half of



March. Overall, the ratios of monoterpenes/isoprene agree with the values reported for a topical forest region in SE Asia. The estimated contribution from local biogenic sources to ambient monoterpenes increased from 31% in first half of February to 67% in second half of March. This trend suggests the increasing biogenic contribution from February to March. The NW winds and higher ambient temperatures in March favored the local emissions and regional transport of BVOCs.

**Keywords:** Monoterpenes; Biogenic emissions; VOCs, Winter-Summer Transition, Tropical Asia, India

# 1. Introduction

In the atmosphere, photo-oxidation reactions of volatile organic compounds (VOCs) and oxides of
nitrogen (NOx =NO+NO2) lead to the production ozone ($O_3$) (Liu et al., 2008). Secondary organic aerosol (SOA) is produced from gas-to-particle conversion processes of VOCs released into the atmosphere from anthropogenic and biogenic sources (Jimenez et al., 2009). Elevated levels of VOCs can suppress the concentration of hydroxyl (OH) radical and enhance the formation of peroxy ($HO_2$ and $RO_2$) radicals and organic nitrates such as peroxyacetyl nitrate (PAN) (Stewart et al., 2003). Globally,
VOCs are emitted from various anthropogenic and natural sources. Major sources of anthropogenic VOCs (AVOCs) include combustion of fossil fuel, biomass burning, use of solvents, industrial production, refineries, etc.(Sahu, 2012). The terrestrial vegetations, particularly in the tropics, emit large quantities of biogenic volatile organic compounds (BVOCs) including reactive non-methane hydrocarbons (NMHCs) and oxygenated-VOCs (OVOCs). Terpenoids such as isoprene ($C_5H_8$),
monoterpenes ($C_{10}H_{16}$) and sesquiterpenes ($C_{15}H_{24}$) are the primary constituents of BVOCs. Atmospheric photochemical oxidation of monoterpenes particularly ozonolysis reactions produce SOA (Haase et al., 2011). Many plants emit BVOCs due to normal metabolism and stress metabolism (Fall, 1999; Peñuelas and Staudt, 2010). The emission flux of BVOCs from plants depends on environmental condition (Calfapietra et al., 2013; Loreto and Schnitzler, 2010). For example, emission rates of BVOCs



are generally dependent upon solar radiation, temperature, relative humidity (RH) and concentration of carbon dioxide ($CO_2$) leading to substantial diurnal variability (Kesselmeier and Staudt, 1999; Portillo‑Estrada et al., 2018). Monoterpene emissions vary significantly between different plant species so the vegetation type has a large impact on total BVOC emissions from a region. The model parameterizations of monoterpene emission are based on normal or average weather conditions which

do not necessarily take into account of change in formation and release rates under extreme and episodic changes in weather conditions (Haase et al., 2011). In plants, storage structures such as the resin duct and the glandular cells lead to pool emissions and account for both the daytime and nighttime release of terpenes. The emission of terpenes from recently assimilated carbon and which is tightly coupled with photosynthesis and metabolism is another important pathway (Wu et al., 2017). Emission rates of

monoterpenes are also influenced by age of leaf and tree (Thoss et al., 2007) water and nutrient availability (Blanch et al., 2009) and seasonality (Hakola et al., 2006). Both isoprene and monoterpenes are highly reactive species and have large and varying effects on regional atmospheric chemistry (Hallquist et al., 2009). Monoterpenes have a short atmospheric lifetime ranging from several minutes to hours due to rapid reactions with OH, $O_3$ and nitrate ($NO_3$) (Atkinson and Arey, 2003).

Global emission fluxes of BVOCs are estimated to be ~7 times greater than those of AVOCs (Stewart et al., 2003). The annual global emissions of BVOCs have been estimated to be about 1150 TgC $yr^{-1}$ comprising about 44% of isoprene and 11% of monoterpenes (Guenther et al., 1995, 2012; Sindelarova et al., 2014). The emissions from tropical region contribute about 80-85% to the global budgets of isoprene and monoterpenes (Sindelarova et al., 2014). But there are considerable uncertainties in global

emission estimates due to the large number of BVOC species and a variety of biological sources (Guenther, 2013). In the literature, widely diverging emission estimates of monoterpenes are due to the lack of observational data and poor understanding of emission mechanism (Arneth et al., 2008). Among several monoterpenes the most dominant species in ambient are $\alpha$-pinene and $\beta$-pinene (Jardine et al., 2015). Much of the recent works on emission and atmospheric chemistry of BVOCs have been focused

on isoprene. Emission rates of monoterpenes are estimated to be small compared to isoprene but it can have a disproportionate effect on aerosol formation due to higher SOA yields (Griffin et al., 1999). The



need of regional representations of BVOCs, particularly of monoterpenes, has been emphasized in the state-of-art models of atmospheric aerosols (Browne et al., 2014). With the exception of isoprene, the chemical scheme for SOA formation is not coupled in the global models such as the Goddard Earth
Observing System Chemistry Climate Model (GEOS CCM).

Anthropogenic emissions are a dominant source of VOCs in urban and industrial regions but BVOCs have also been reported to make a significant contribution to total VOC in many cities of the world (Calfapietra et al., 2013 and references therein). In the tropics, sparse measurements of BVOCs cause higher uncertainties in their emission estimates and atmospheric importance. Although there has been
significant growth of literature available about tropical BVOC emission in Amazonia and in tropical Africa, but there is still a paucity of data on BVOC emission in tropical Asia (Wang et al., 2007). The measurements of BVOC emission rates from tropical plants are limited and are particularly lacking over the Indian subcontinent (Varshney and Singh, 2003). In India, emissions of isoprene from common plant species have been examined to some extent (Singh et al., 2011; Varshney and Singh, 2003). The present
study is based on analysis of ambient mixing ratio data of monoterpenes measured using proton transfer reaction time of flight mass spectrometer (PTR-TOF-MS) instrument at an urban site of Ahmedabad in western India during February-March 2014. The main objective of this study is to examine the temporal trends and assess impact of meteorological parameters in context of the transition from winter-to-summer conditions.


## 2. Measurement site and PTR-TOF-MS instrumentation

The measurement site (23.0356° N; 72.5435° E, 49 m above sea level) is located in the campus of Physical Research Laboratory (PRL) in Ahmedabad city. It is the largest city in the state (province) of Gujarat with a population of more than 6.3 million and area of about 466 km$^2$. The *Sabarmati* river
divides the city into eastern and western parts. As shown in Fig. 1, the study site is located in the western part of the city where vehicle exhaust is a major source of air pollution including AVOCs (Sahu





et al., 2016a). According to latest data, about 3.2 million vehicles are registered in Ahmedabad which is increasing at a rate of about 10% yr$^{-1}$ (Chandra et al., 2016). The eastern part has several industrial estates especially of textiles, chemicals, plastics, metal alloys, machinery, etc. As shown in Fig. 1, the inner and outer roads encircling the city are the 132 feet ring road and Sardar Patel (SP) ring road, respectively. Therefore, irrespective of wind direction, emissions from vehicle exhaust are expected to be the dominant source of AVOCs in the city. The trees cover about 6% of total area in Ahmedabad city. About 6.18×10$^5$ trees were counted in the year 2011 with relatively higher coverage in the western and south parts than those in eastern and north parts. As shown in Fig. 1, the major plant species found in the city are *Azadirachta indica*, *Polyalthia longifolia* and *Holoptelea integrifolia* and *Delonix regia* (Limbochiya and Patel, 2013). The significant emissions of monoterpenes from the major plant such as *Azadirachta indica* and *Delonix regia* are reported in forested areas of *Haryana* state and Delhi in India (Padhy and Varshney, 2005; Singh et al., 2011). The list of trees also includes some high monoterpene emitters such as *Mangifera indica*, *Syzygium cumini* and *Eucalyptus globulus*. The surrounding areas of the city are mostly agricultural land with a lower tree cover of about 4.75% of the total geographical area. The *Thol* wildlife sanctuary with a total area of 700 ha is situated at a distance of about 25 km northwest (NW) of Ahmedabad city. The vegetations found in the *Thol* sanctuary are mainly scrub type with mixed flora of aquatic and marshy plants (Vyas, 2014). Gandhinagar, also known as India's tree capital which has a green cover of about 54%, is located approximately 23 km north of Ahmedabad city. Therefore, relatively higher levels of BVOCs are expected during February-March period when NW winds prevail.

Real-time continuous measurements of ambient VOCs were made using the PTR-TOF-MS 8000 instrument (Ionicon Analytik GmbH, Innsbruck, Austria) during 01 February-31 March 2014. The PTR-TOF-MS 8000 instrument provides very fast, sensitive and high mass resolution measurements of various VOC compounds present in the air. The drift tube pressure of ~2.3 mbar, the drift tube temperature of 60 °C and the drift tube voltage of 660 V were maintained throughout the study period. This drift tube setup corresponds to an *E*/*N* ratio of 128-130 Townsend (Td) (1 Td = 10$^{-17}$ V cm$^2$), where *E* is electric field and *N* is number gas density. The typical *E*/*N* ratios of 120-140 Td is recommended for



ground level ambient measurements using the PTR-MS (Hewitt et al., 2003). Ambient air flow rate of

130 60 mL min$^{-1}$ through a 1.5 m long heated (60 °C) Teflon® PFA tube (1/8" or 3.175 mm) was maintained

during the study period. The raw data files in 'hdf5' format were acquired using the TOF-Daq software

(version 1.2.93, Tofwerk AG, Switzerland). Subsequently, the "PTR-TOF Data Analyzer" software was

used for further processing of the data (Müller et al., 2013). In this paper, we have presented the mixing

ratio data of monoterpenes (MTs) measured at $m/z$ 137.131 ($C_{10}H_{16}$-H$^+$). A certified gas mixture

135 containing 0.97 ppmv ± 5% of $\alpha$-pinene (L5388, Ionicon Analytik GmbH Innsbruck) was used for the

calibration and determination of sensitivity. The measurement precision of 2.7% was determined using

20 min integration of 30 s time resolution data at a constant value of about 3 ppbv. The overall accuracy

of monoterpene data is estimated to be about 10% which is mainly due to the uncertainties in standard

mixture and set flow rates of gases in the gas calibration unit (GCU). More details of PTR-TOF-MS,

operational setup, data processing and calibration are given in our previous study (Sahu et al., 2016b). In

this study, we have also used ambient mixing ratios of other VOCs such as benzene and isoprene as

supporting data reported in our previous paper (Sahu et al., 2017).

## 3. Results and discussion

### 3.1 Time series of monoterpenes and meteorological parameters

Hourly and daily time series of monoterpenes mixing ratio show large variations during the study period

(Fig. 2). In February, hourly average mixing ratios varied in the range of 0.05-3.5 ppbv, while daily

averages were in the range of 0.11-0.92 ppbv. The monthly average mixing ratio of monoterpenes in

February was 0.37±0.2 ppbv. In short time scales (<24 h), large periodic variations indicate strong

diurnal dependence of monoterpenes, while day-to-day variations seem to be controlled by the synoptic-

scale weather conditions. Time series of wind speed, temperature, pressure and solar flux and planetary

boundary layer (PBL) depth are also plotted in Fig. 2. Typically, the periods of low and high mixing

ratios coincide with the episodes of strong and weak winds, respectively. For example, daily mean

values exceeding 0.5 ppbv during 2-5 February and 18-20 February coincided with light winds and



warmer weather conditions with daily maximum temperature exceeding 30$^{o}$C. On the other hand,

periods of low monoterpenes (<0.2 ppbv on daily scale) during 10-13 February and 23-24 February were influenced by relatively strong winds and lower surface temperatures. The actual concentration depends on the emission flux, oxidation and vertical mixing (Mielke et al., 2010). Therefore, signatures of biogenic emissions in the mixing ratio data are obscured due to dominating influence of vertical mixing and oxidation loss processes. However, to some extent, the ratio of monoterpenes to benzene (an

anthropogenic tracer) can take account of variations due to change in local meteorology and PBL. Hourly monoterpenes/benzene ratio exhibits large periodic variation which tends to follow the diurnal cycle of temperature. Monoterpenes/benzene ratio showed slightly increasing trend with average values of 0.19±0.03 and 0.26±0.07 ppbv ppbv$^{-1}$ during first and second halves of February, respectively.

Time series of monoterpenes in March also showed significant hourly and day-to-day variations. The

amplitudes of hourly data during the first half of March were larger than those observed during the second half of March. In this month, hourly and daily mean mixing ratios of monoterpenes varied in the ranges of 0.06-6.1 ppbv and 0.09-1.12 ppbv, respectively. The monthly average mixing ratio of monoterpenes was 0.35±0.3 ppbv in March. The periods of elevated monoterpenes, for example during 15-17 March, coincided with calm winds and warmer temperatures. Hourly data of

monoterpenes/benzene ratio exhibit large periodic variations and tends to follow cycle of ambient temperature on diurnal scale. Daily averages of monoterpenes/benzene ratio were in the range of 0.11-1.68 ppbv ppbv$^{-1}$ during March. Most significantly, the ratio of monoterpenes/benzene increased from 0.20±0.07 ppbv ppbv$^{-1}$ in the first half of February to 0.50±0.40 ppbv ppbv$^{-1}$ in the second half of March. Monthly average of monoterpenes/benzene ratio in March (0.36±0.16 ppbv ppbv$^{-1}$) is

significantly higher than that in February (0.22±0.06 ppbv ppbv$^{-1}$). This trend in monoterpenes/benzene ratio is consistent with that of daily mean temperature which increased from 17-25 $^{o}$C in February to 20-30 $^{o}$C in March. Statistics of monoterpenes, monoterpenes/benzene ratio and meteorological parameters for four different periods of February-March, 2014 are presented in Table 1. The mean mixing ratios of monoterpenes in rural New Hampshire, USA were 0.50 ppbv in the summer season and 0.10 ppbv in the

winter season (Haase et al., 2011). The monthly daytime and nocturnal average mixing ratios of at New





Hampshire were ~0.6 and 1.6 ppbv, respectively. At a site in Montseny, NE Spain, levels of monoterpenes were 2.56 ppbv and 0.23 ppbv during the summer and winter seasons, respectively (Seco et al., 2011).

## 3.2 Impact of biomass burning

The major anthropogenic sources of monoterpenes are wood processing in sawmill, biomass burning burning and traffic (Hellén et al., 2012; Koss et al., 2018; Schade and Goldstein, 2003). Several studies have reported that the massive amounts of BVOCs can be emitted in response to burning vegetation and wounding at high temperatures (Loreto and Schnitzler, 2010; Müller et al., 2016). Among different BVOCs, emission of α-pinene has been found as a major monoterpene in fresh biomass burning plumes (Akagi et al., 2013). The mixing ratio of acetonitrile ($CH_3CN$) is commonly used as a marker to track biomass burning plumes (Holzinger et al., 1999). The time series of MTs and acetonitrile and their correlations during two major events of biomass burning are shown in Fig. 3. In the first episode, from the evening of 19 February until the morning of 20 February, the mixing ratios of monoterpenes and acetonitrile show large simultaneous enhancements with their peak values exceeding 3 ppbv (Fig. 3a). The strong correlation ($r^2$ = 0.91) and a high$\Delta$MTs/$\Delta$CH $_3$CN slope value of 1.08 ppbv ppbv$^{-1}$ indicate strong impact of fresh biomass burning emissions (Fig. 3b). In Fig. 3 c and d, variation of monoterpenes and acetonitrile are shown for another major event of biomass burning from the evening of 15 March until early morning of 16 February coincide with *Holi* festival bonfire in India. *Holi* is a spring festival in India which is celebrated in the month of *Phalgun* according to the Indian Calendar or in the months of February/March. Traditionally, huge bonfires are lit on the night preceding *Holi* festival and the event is known as '*Holika Dahan*'. Therefore, huge amounts of semi-dry biomass and wood burned across the Indian subcontinent could contribute to the emission of many trace gases including BVOCs. During *Holika Dahan,* the mixing ratios of monoterpenes and acetonitrile show large and correlated enhancements with their peak values exceeding 6 ppbv and 4 ppbv, respectively (Fig. 3c). The strong



correlation ($r^2 = 0.89$) and a high $\Delta MTs/\Delta CH_3CN$ slope value of 1.46 ppbv ppbv$^{-1}$ indicate strong impact of the *Holi* bonfire (Fig. 3d).

## 3.3 Diurnal profiles of monoterpenes and monoterpenes/benzene ratio

Average diurnal mixing ratio profiles of monoterpenes for four different periods of 1-15 February, 16-28 February, 1-15 March and 16-31 March are plotted in Fig. 4a. Mixing ratio of monoterpenes showed strong diurnal dependence throughout the study period. Except for the afternoon hours, mixing ratios of monoterpenes showed significant differences between the first and second halves of February. From night till early morning hours, mixing ratios of monoterpenes observed in the second half of February

were about 2 times higher than those in the first half. Mixing ratios were moderate from midnight till the noon (00-12 h) and decreased to the lowest values in the afternoon (12-17 h). The afternoon values increased slightly from 0.11±0.02 ppbv in the first half of February to 0.16±0.01 ppbv in the second half of March. From evening till midnight (18-24 h), mixing ratios (0.20-0.75 ppbv) were elevated but exhibit significant differences between the four different periods. The summer season diurnal pattern of

monoterpenes in Montseny (Spain) showed higher daytime values of ~1.6 ppbv and lower night values of ~0.2 ppbv (Seco et al., 2011). Warneke et al., (2004) report that the highest mixing ratios of 0.34 ppbv during nighttime and lowest at midday (0.02 ppbv) at New England in summer season.

Several local factors such as emission, photo-oxidation and PBL processes play important role in controlling the diurnal dependence of monoterpenes. Biogenic emissions of monoterpenes occur

throughout the day and night but their strong diurnal dependence indicates buildup in the shallower nighttime boundary layer (Kaser et al., 2013). The combined effect of growing PBL depth and reactivity with $O_3$ and OH leads to depletion during daytime (Hakola et al., 2012). Therefore, biogenic emissions which primarily occur during the daytime seem not enough to counter the effects of removal due to dynamical and chemical processes in the lower troposphere. Although qualitatively, ratio of

monoterpenes/benzene may take account of the variations caused by local meteorology and PBL dynamics. But photochemical reaction rates with OH for monoterpenes (6.08-7.72×10$^{-11}$ cm$^3$ molecule$^{-1}$



$s^{-1}$) and benzene ($1.22 \times 10^{-12}$ cm$^3$ molecule$^{-1}$ s$^{-1}$) differ significantly (Atkinson and Arey, 2003; Chuong et al., 2002). Therefore, the ratio of monoterpenes/benzene does not fully take account of photochemical loss but underestimates the enhancement caused by biogenic emissions. Nonetheless, the purpose of this study is to demonstrate the relative change in biogenic emissions due to change in environmental parameters associated with winter-to-summer transition.

Average diurnal profiles of monoterpenes/benzene ratio can be described as bimodal with peaks in the early morning and afternoon hours (Fig. 4b). Although the ratio showed similar diurnal patterns during different periods but the levels increased rapidly from winter to summer. The lowest values (0.15-0.20 ppbv ppbv$^{-1}$) were observed from evening till midnight during each of four periods. While elevated ratios were observed in the afternoon hours (13-18 h) with peak at around 16 h. In the afternoon, the ratio increased from 0.17±0.03 ppbv ppbv$^{-1}$ in the first half of February to 0.43±0.07 ppbv ppbv$^{-1}$ in the second half of March. In the afternoon hours, enhancements of monoterpenes/benzene ratio coincided with the highest temperature. While enhancements during the early morning hours (00-05 h) coincided with lowest temperature of the day. In contrast to isoprene, emissions of monoterpene are independent of light but exponentially dependent on ambient temperature (Mielke, 2010). Therefore, in the absence of sunlight, higher MTs/isoprene ratios indicate higher nighttime biogenic emissions of monoterpenes than those of isoprene. The lower ratios of MTs/isoprene persisted throughout the day suggest that the high radiation favored the isoprene emissions relative to monoterpenes (Ebben et al., 2012). During the midnight-early morning period, the average MTs/isoprene ratio of 0.25 ppbv ppbv$^{-1}$ in the first half of February increased to 0.43 ppbv ppbv$^{-1}$ in the second half of March (Fig 4c). The estimated change in MTs/isoprene ratio is consistent as the change of ambient temperature in the early morning hours with significantly higher values in second half of March than those in the first half of February. The nighttime emission of monoterpenes would have caused the increase in ambient mixing ratio due to the development of the nocturnal boundary layer (NBL). Therefore, the nighttime enhancements of monoterpenes/benzene ratio indicate the emissions from non-specific storage pool (Demarcke et al., 2010). It has been reported that the plants accumulate pools of monoterpenes and store them in structures like resin ducts and glandular trichomes or related structures (Kuhn et al., 2002). The emission





of isoprene generally depends upon both light and temperature, whilst release of monoterpenes from plant storages may be exclusively by temperature and independent of light (Jones et al., 2011). In the afternoon and early morning hours, the ratios of MTs/isoprene were in the ranges of 0.1-0.2 ppbv ppbv$^{-1}$ and of 0.25-0.43 ppbv ppbv$^{-1}$, respectively. In the tropical rainforest of South-East Asian (SE Asian), the emission ratios of MTs/isoprene were measured in the range of 0.24-0.27 (Misztal et al., 2011). The daytime MTs/isoprene ratios measured in the present study are in good agreement with the average ratios of 0.23±0.3 derived from flux measurements at the Bukit Atur GAW station in SE Asian tropical rainforest (Langford et al., 2010). In fact, the MTs/isoprene ratio increased from ~0.3 during the day to ~2.0 during the night at Bukit Atur station (Jone et al., 2011). Guenther et al. (2008) report the MTs/isoprene emission ratios of ~0.15 for different tropical forests. The increasing trend of monoterpenes/benzene ratio in the afternoon hours is consistent with the gradual increase in ambient temperature during winter to summer transition. The average diurnal maximum of solar flux (temperature) increased from 667 W m$^{-2}$ (30 $^o$C) in the first half of February to 767 W m$^{-2}$ (38 $^o$C) in the second half of March.

In Fig. 4e-h, diurnal plots of monoterpenes mixing ratio, monoterpenes/benzene ratio, ambient temperature and solar flux for clear-sky and cloudy days are compared to evaluate the sensitivity of biogenic emissions. On a cloudy day (15 February), mixing ratio of monoterpenes exhibited large variation and higher values compared to a clear-sky day (17 March). While monoterpenes/benzene ratio showed opposite patterns to that of diurnal pattern of monoterpenes mixing ratio. In the afternoon hours, ratios of monoterpenes/benzene on a clear-sky day were significantly higher than those observed on a cloudy day. The ratios of monoterpenes/benzene for cloudy and clearly-sky days were 0.21±0.08 ppbv ppbv$^{-1}$ and 0.60±0.22 ppbv ppbv$^{-1}$, respectively. Except some spikes, ratios of monoterpenes//benzene followed the diurnal pattern of ambient temperature on both cloudy and clearly-sky days.

## 4. Effect of local meteorology

## 4.1 Dependence on wind parameters





Meteorological parameters play a vital role in dispersion and accumulation of pollutants including short-lived VOCs in urban regions. Meteorological conditions can also influence biogenic emission processes through mechanical stress to plants and hence the mixing ratio of monoterpenes (Haase et al., 2011). The relationships of monoterpenes mixing ratio with wind speed in February and March are shown in Fig. 5. Box-whisker plots using 10-minute interval data show the levels of different percentiles (5, 25,

75 and 95), mean and median in each wind speed bin (0.5 m s$^{-1}$). Mixing ratio of monoterpenes and its variability declined in the lower wind speed regimes (<3 m s$^{-1}$), while showed little response at higher wind speeds (>3 m s$^{-1}$). In February (March), average mixing ratios were 0.75±0.84 ppbv (0.52±0.84 ppbv) and 0.20±0.84 ppbv (0.15±0.09 ppbv) under calm and windy conditions, respectively. The dependence of monoterpenes mixing ratio with wind speed can be represented by exponential decay

functions.

$$\text{In February: } [\text{MTs}] = 0.661 \times \exp(-0.985 \times WS) + 0.166 \qquad (1)$$

$$\text{In March: } \quad [\text{MTs}] = 0.493 \times \exp(-0.496 \times WS) + 0.088 \qquad (2)$$

In above equations, [MTs] is mixing ratio (ppbv) of monoterpenes and WS stands for wind speed (m s$^{-1}$). Comparison of fit parameters in Eq. (1) and (2) indicates that the decline rate of monoterpenes with

increasing wind speed in February was much faster than that in March.

The ratio of monoterpenes/benzene increased gradually from 0.20±0.25 ppbv ppbv$^{-1}$ under calm winds to 0.35±0.30 ppbv ppbv$^{-1}$ at higher winds (5-6 m s$^{-1}$) in February. In March, the ratio showed a different relationship with wind speed as it increased rapidly from 0.25 to 0.90 ppbv ppbv$^{-1}$ at lower wind speeds but showed sharp decline to 0.15±0.09 ppbv ppbv$^{-1}$ at higher wind speeds. The distinct relations between

monoterpenes/benzene ratio and wind speed in February and March suggest the significant roles of other factors. For example, unlike February, the higher wind speeds in March coincided with lower ambient temperatures (mostly below 30 $^{o}$C). Consequently, declines of monoterpenes/benzene ratio at higher wind speeds in March could be due to reduced biogenic emissions. Similarly, highlighting the impact of



wind speed Hasse et al. (2011) have reported a factor of 93% increase of monoterpenes mixing ratio
during storms at the Thompson Farm site in Durham, New Hampshire.

To assess the wind direction dependence, we have used a polar frequency analysis of monoterpenes
mixing ratio and monoterpenes/benzene ratio (Fig. 6). The occurrence frequencies of different levels of
monoterpenes and monoterpenes/benzene are averaged in a bin of 22.5°. Winds were predominantly
from the N-NE during both the months but presence of W-NW winds was also significant in March. In
February (March), elevated mixing ratios of monoterpenes (>0.5 ppbv) and monoterpenes/benzene
ratios (>0.5 ppbv ppbv$^{-1}$) account for about 25% (18%) and 4% (14%) of total data, respectively. In
other words, from winter-to-summer transition, mixing ratio of monoterpenes and ratio of
monoterpenes/benzene showed decreasing and increasing trends, respectively. In particular, the highest
frequency of elevated monoterpenes was observed during northerly winds in February while during
westerly winds in March. Overall, the elevated mixing ratios of monoterpenes were mostly observed
during easterly (0-180°) winds in February, while during westerly (180-360°) winds in March. Mixing
ratios of monoterpenes were about 0.35 ppbv during both easterly and westerly winds. However
interestingly, the ratios of monoterpenes/benzene differed significantly with average values of 0.21 ppbv
ppbv$^{-1}$ in the easterly and 0.49 ppbv ppbv$^{-1}$ in the westerly winds. The wind direction dependence of
monoterpenes/benzene ratio clearly indicates higher contributions from biogenic sources located in the
western regions. Therefore, the combined effect of westerly winds and higher ambient temperatures in
March provides favorable conditions for the local emissions and regional transport of BVOCs.

Photo-oxidation of monoterpenes leads to the formation of SOA and hence can cause significant
reductions in atmospheric visibility. Atmospheric visibility in urban areas has been observed to show
positive correlations with wind speed and ambient temperature (Chen and Xie, 2013). We have analyzed
the variation of monoterpenes mixing ratio and monoterpenes/benzene ratio with visibility (Fig. 7).
Mixing ratios of monoterpenes were high in the lower visibility conditions (<4 km) but showed lower
values under high visibility conditions (>4 km). In February (March), the mixing ratio of monoterpenes
decreased from 0.58±0.4 (0.47±0.22) ppbv at 1.5 km of visibility to 0.20±0.2 (0.15±1.2) ppbv at 5.5 km.





While the ratio of monoterpenes/benzene tends to increase with increasing visibility. In February (March), the monoterpenes/benzene ratio increased from 0.0.11±0.01 (0.19±0.07) ppbv at 1.5 km of visibility to 0.24±0.3 (0.30±2.4) ppbv at 5.5 km.

## 4.2 Dependence on ambient temperature and sunlight


Ambient temperature is one of the most important parameters controlling emission of BVOCs. Variations of monoterpenes mixing ratio and monoterpenes/benzene ratio with ambient temperature are shown in Fig. 8. Box plots include 10-min interval data measured during daytime in a bin of 2.0 $^o$C. The ratio of monoterpenes/benzene exhibits little dependence while mixing ratio of monoterpenes tends to

decrease with increasing temperature in February. In this month, the ratio of monoterpenes/benzene was slightly low (0.24±0.2 ppbv ppbv$^{-1}$) at 11-25 $^o$C and high (0.29±0.2 ppbv ppbv$^{-1}$) at 25-35 $^o$C. In March, the ratio gradually increased from 0.27±0.2 ppbv ppbv$^{-1}$ at lower temperatures to 0.50±1.5 ppbv ppbv$^{-1}$ at higher temperatures. Unlike those in February, the daytime mixing ratio of monoterpenes shows sharp rise at higher temperatures (>30 $^o$C) in March. In this month, the daytime mixing ratios of monoterpenes

were 0.17±0.02 ppbv at lower temperatures (<30 $^o$C) and 0.35±0.2 ppbv at higher temperatures (>30 $^o$C). Sunlight intensity is another significant factor controlling emission of BVOCs from vegetation. Variations of monoterpenes mixing ratio and monoterpenes/benzene ratio with sunlight intensity are also shown in Fig. 8. In February, the ratio decreased slightly with increasing solar flux while mixing ratio of monoterpenes does not show a clear trend. In March, both monoterpenes mixing ratio and

monoterpenes/benzene ratio increased rapidly at lower intensities (<150 W m$^{-2}$) but show no clear dependence at higher intensities. Quantitatively, the dependence of monoterpenes and monoterpenes/benzene on ambient temperature is complicated due to several other competing and simultaneous processes. For example, emissions are high at higher temperatures, at the same time removal rates due to photo-oxidation and PBL dynamics are also high. Despite this complexity,

significant increase in biogenic emissions of monoterpenes is due to change from winter to summer



conditions. In monoterpene emitting vegetations, monoterpenes are stored in storage structures referred as pool including leaf cavities, glandular cells, resin canals and ducts. As a result, monoterpene emissions are mainly dependent on ambient temperature (Guenther et al., 2012).

## 4.3 Estimation of biogenic emissions and impact of other sources

Time series of monoterpenes mixing ratio and monoterpenes/benzene ratio for the day and night periods are separately shown in Fig. 9. The nighttime mixing ratios exhibit strong variations in the range of 0.10-1.0 ppbv compared to daytime values of 0.08-0.48 ppbv. The daytime mixing ratios were consistently lower than the nighttime values from February until the first half of March but were comparable during the second half of March. In case of monoterpenes/benzene ratio, the daytime (0.21 ppbv ppbv$^{-1}$) and nighttime (0.19 ppbv ppbv$^{-1}$) values were comparable from February until the first half of March. However, in the second half of March, the daytime ratio of 0.48 ppbv ppbv$^{-1}$ was significantly higher than the nighttime ratio of 0.28 ppbv ppbv$^{-1}$. Overall, the daytime mixing ratio of monoterpenes and ratio of monoterpenes/benzene follow the trend of ambient temperature. The enhancements in the daytime monoterpenes mixing ratio and monoterpenes/benzene ratio coincide with the periods of higher temperatures when daily maximum exceeded 30 $^{\circ}$C. Which were particularly noticeable when winds were predominantly from the W-N sector (270-360$^{\circ}$). Overall, daytime and nighttime mixing ratios of monoterpenes showed increasing and decreasing trends, respectively. In particular, very high nighttime mixing ratios were measured during periods of shallower PBL depths. But monoterpenes/benzene ratio during both the periods of day and night showed increased trends.

A source-tracer-ratio method has been applied to estimate the contributions of local and regional sources to the abundance of monoterpenes (Legreid et al., 2007). The ambient mixing ratio of monoterpenes (MT) can be represented as the sum of contributions of primary anthropogenic (MT$_A$), biogenic (MT$_B$) and transport (MT$_R$) sources.

$$MT = MT_A + MT_B + MT_R \qquad (3)$$



In Eq. (3), $MT_A$ represents local contribution mainly from biomass/bio-fuel burning. During present study period, $MT_R$ could represent contributions from both biomass burning and biogenic emission that have taken place at far distances. However, due to relatively short atmospheric lifetime of monoterpenes, values of $MT_R$ are expected to be small or at least assumed to be constant during the study period (Sahu et al., 2017). Anthropogenic contribution can be estimated by using benzene as a marker (Wagner and Kuttler, 2014).

$$MT_A = (\Delta MT/\Delta benzene) \times \text{Mixing ratio of Benzene} \qquad (4)$$

Where, $\Delta MT/\Delta benzene$ is emission ratio (ER) estimated from the evening measurements influenced mainly by local anthropogenic sources and have not been significantly altered due to oxidation processes. Nonetheless, activities of biomass/bio-fuel burning in the city and nearby areas were negligible. Therefore, relative contribution of biogenic sources to monoterpenes mixing ratio can be estimated as here.

$$MT_B = MT - (\Delta MT/\Delta benzene) \times \text{Mixing ratio of Benzene} \qquad (5)$$

The average contributions from local biogenic sources to ambient monoterpenes were estimated to about 31%, 49%, 41% and 67% during 1-15 February, 16-28 February, 1-15 March and 16-31 March, respectively. Temporal trend of biogenic contribution follows the variation of daily maximum temperature to large extents. At the same time, it is important to take account of the fact that the biogenic contributions were lowest and highest during the winds from the N-E and W-N sectors, respectively. Therefore, the combined effect of ambient temperature and wind direction explains the overall trend of biogenic contribution. As reported in Sahu *et al*. (2017) that the mixing ratio of acetonitrile does not show any clear trend during the winter-summer period. Therefore, the estimated trend of biogenic contribution is expected to have little influence of biomass/bio-fuel burning. The increase of about 36% from winter to summer conditions can be attributed to the change in biogenic emissions. This is more than 2 times higher than estimated that of isoprene (16%) during the same study





period (Sahu et al., 2017). Accurate measurement of BVOCs is important due to significant effects chemical compositions of the tropical troposphere.

In addition to direct emissions from anthropogenic and biogenic sources ambient OVOCs are formed by photo-oxidation of precursor gases (Sommariva et al., 2011). In urban regions, monoterpenes and some other NMHCs are major precursors of secondary OVOCs. Among BVOCs, monoterpenes are also

considered to be one of the major precursors of photochemical formation of acetone. Photochemical production from the oxidation of monoterpenes contributes about 64% to the global acetone sources (Khan et al., 2015). In this study, ratio of acetone/benzene has been considered to take account of primary anthropogenic emissions and variation due to PBL dynamics. Scatter plots between acetone/benzene ratio and monoterpenes mixing ratio for months of February and March are shown in

Fig. 10. The data points show clear segregation depending on the local time of measurements in both the months. Most of the data points measured in the daytime are populated along the high acetone/benzene ratios and low monoterpenes, while vice versa during night and morning hours. In the daytime, ratio of acetone/benzene exhibited rapid increase with decreasing mixing ratio of monoterpenes during both the months. However, this tendency was stronger in March than that in February. The median ratios of

acetone/benzene were 6.6±7.5 and 8.2±8.0 ppbv ppbv$^{-1}$ at lower values (<0.2 ppbv) of monoterpenes which decreased to 1.2±0.45 and 1.7±0.25 ppbv ppbv$^{-1}$ at higher values (>1.0 ppbv) in February and March, respectively. In the daytime, higher acetone/benzene ratios in March compare to those in February indicate higher contributions from both photo-oxidation and local biogenic sources. Despite significant biogenic emissions, lower daytime values of monoterpenes indicate significant

photochemical loss which possibly leading to the production of OVOCs including acetone. Higher daytime ratios of acetone/benzene in March compare to those in February indicate more contributions from photo-oxidation and biogenic sources. Fischbeck et al. (2017) reported a rapid increase in biogenic emission and secondary formation of acetone during winter-summer transition period over South Asia region. Scatter plots between mixing ratios of monoterpenes and acetonitrile are shown in Fig. 10. In the

daytime, the change in mixing ratio of monoterpenes does not show any clear dependence on acetonitrile. In fact, large enhancements of monoterpenes can be noticed at lower acetonitrile



concentrations. The correlated increases of both compounds during the night are due to regional sources. In shallower NBL, different trace gases show good correlation even though they are not emitted from same source. Therefore, the estimated trend of monoterpenes is mainly governed by the local biogenic
sources and other sources contribute to regional background.

## 5. Conclusions

Variation of monoterpenes and impact of meteorological parameters were examined in the context of the transition from winter-to-summer at an urban site of India. Mixing ratios of monoterpenes ($m/z$ 137.131)
were measured using PTR-TOF-MS instrument during February-March 2014. The data presented in this study is one of the best available measurements in India owing to high time- and mass- resolutions of PTR-TOF-MS. The meteorological parameters showed significant changes with prevailing winds gradually shifted from northerly in February to westerly in March. Large increases in monoterpenes are observed under warm and calm conditions. Mixing ratio of monoterpenes showed strong diurnal
dependence with elevated values from evening till midnight hours and lowest in the afternoon. Some sporadic local biomass burning events including the bonfire during *Holi* festival caused exceptionally high levels of monoterpenes in ambient air.  Daily mixing ratio of monoterpenes does not show clear trend with monthly means of about 0.35 ppbv for both the months. The impact of biogenic emissions could not be discerned in mixing ratio data due to competing influence of PBL dynamics and OH-
reaction loss processes. However, monoterpenes/benzene ratio increased from 0.22±0.06 ppbv ppbv$^{-1}$ in February to 0.36±0.16 ppbv ppbv$^{-1}$ in March associated with increasing ambient temperature. In the afternoon, average monoterpenes/benzene ratio of 0.43 ppbv ppbv$^{-1}$ measured in the second half of March was about 2 times higher than that in the first half of February. Mixing ratio of monoterpenes declined with increasing wind speed which is well represented by the exponential decay functions.
However, rate of decline in February was about 2 times greater than that in March. Monoterpenes/benzene ratio shows strong response with ambient temperature as it increased from 0.27 ppbv ppbv$^{-1}$ ($<30^{\circ}$C) to 0.50 ppbv ppbv$^{-1}$ ($>30^{\circ}$C). The increase of about 36% in biogenic contributions

of monoterpenes during winter-to-summer conditions can be attributed mainly to local sources. The higher nighttime ratios of monoterpenes/isoprene indicate significant light-independent but temperature dependent emissions of monoterpenes. Overall, the combined effect of westerly winds and higher temperatures in March provided favorable conditions for the local emissions and regional transport of monoterpenes. This study clearly highlights the strong sensitivity of BVOC emission to the meteorological conditions at semi-arid urban site. At the same time, the inherent complexities due to post-emission processes limit the scope of this data for emission estimates. Reactive BVOCs plays important role in tropospheric chemistry, not only leading to ozone formation but as a precursor for secondary organic aerosol formation. Therefore, flux measurement of BVOCs is highly recommended which will provide emission characteristics in different ecosystems of tropical India. This study highlights the importance of BVOCs considering the policies of improving air quality in tropical urban regions.

*Data availability*. The data used in this study are available in figshare repository (https://figshare.com/s/15ff9a338f800b163c83).

*Author contribution*. In this study, Dr. Lokesh Sahu has designed the experiment. The data analysis was done and also original manuscript was written by Nidhi Tripathi. The corrections and revision in manuscript were made by Dr. Sahu.

*Competing interests*. The coauthor of this article agrees that he has no objection in publishing this work.

*Acknowledgements*. The weather data are taken from wunderground.com. The measurement data presented in this study are available with Dr. Lokesh Kumar Sahu (lokesh@prl.res.in).

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




**Table 1:** Statistics of monoterpenes mixing ratio, monoterpenes/benzene ratio and meteorological parameters measured at Ahmedabad during the four different periods of February-March, 2014.

| **Species** | **01-15 Feb 2014** | **16-28 Feb 2014** | **01-15 Mar 2014** | **15-31 Mar 2014** |
|---|---|---|---|---|
| Monoterpenes (ppbv) | $0.31 \pm 0.19$ | $0.65 \pm 0.45$ | $0.38 \pm 0.22$ | $0.38 \pm 0.26$ |
| Monoterpenes/benzene (ppbv ppbv$^{-1}$) | $0.19 \pm 0.05$ | $0.34 \pm 0.21$ | $0.20 \pm 0.07$ | $0.50 \pm 0.46$ |
| Temperature ($^{o}$C) | $21 \pm 2$ | $22 \pm 2$ | $23 \pm 3$ | $28 \pm 2$ |
| Wind speed (m s$^{-1}$) | $0.27 \pm 0.21$ | $0.82 \pm 0.51$ | $1.29 \pm 0.63$ | $1.13 \pm 0.43$ |
| Pressure (hPa) | $1004 \pm 2$ | $1008 \pm 2$ | $1008 \pm 1$ | $1005 \pm 2$ |
| RH (%) | $52 \pm 6$ | $47 \pm 6$ | $36 \pm 7$ | $38 \pm 5$ |




**Figure captions**

**Figure 1:** Top panel (A) shows the map of India and land use pattern of Ahmedabad city and surrounding areas (prepared using Bhuvan, the ISRO's Earth visualization portal (https://bhuvan.nrsc.gov.in/) and bottom panel (B) shows the number of major trees in Ahmedabad city and emission rates of monoterpenes for different plants taken from Singh et al., (2011) as ER 1 and Malik et al. (2018) as ER 2.


    **Figure 2**: Time series of hourly and daily (box-whisker) monoterpenes (MTs) mixing ratio, MTs/benzene ratio, meteorological parameters and planetary boundary layer (PBL) depth at Ahmedabad from 01 February to 31 March 2014.

**Figure 3:** The time series of MTs and acetonitrile and their correlation plots during two major episodes of biomass burning (a, b) for 19-21 February and (c, d) for 15-16 March, *Holi* festival.

    **Figure 4:** Average diurnal profiles of monoterpenes (MTs) mixing ratio, MTs/benzene ratio, MTs/isoprene ratio, temperature and solar flux during (a-d) four different periods of 1-15 February, 720     16-28 February, 1-15 March and 16-31 March 2014 and (e-h) for a cloudy day (15 February) and clear-sky day (17 March).

    **Figure 5:** Variations of monoterpenes mixing ratio and monoterpenes/benzene ratio with wind speed during the months of February (left panel) and March 2014 (right panel).


    **Figure 6**: Polar frequency (%) plots of monoterpenes mixing ratio and monoterpenes/benzene ratio during the months of February 2014 (left panel) and March 2014 (right panel).

    **Figure 7**: Variation of monoterpenes mixing ratio and monoterpenes(MTs)/benzene ratio with 730     atmospheric visibility in February (left panel) and March 2014 (right panel).

    **Figure 8**: Dependencies of monoterpenes mixing ratio and monoterpenes/benzene ratio with ambient temperature and solar flux during February 2014 (left panel) and March 2014 (right panel).





**Figure 9**: Time series of monoterpenes mixing ratio, monoterpenes/benzene ratio and estimated
       biogenic contribution along with daily air temperatures and percentage occurrence of winds from
       different sectors.

       **Figure 10**: Relations between monoterpenes mixing ratio and acetone/benzene ratio during (a)
February and (b) March 2004 while (c) and (d) are scatter plots between mixing ratios of
       monoterpenes and acetonitrile.






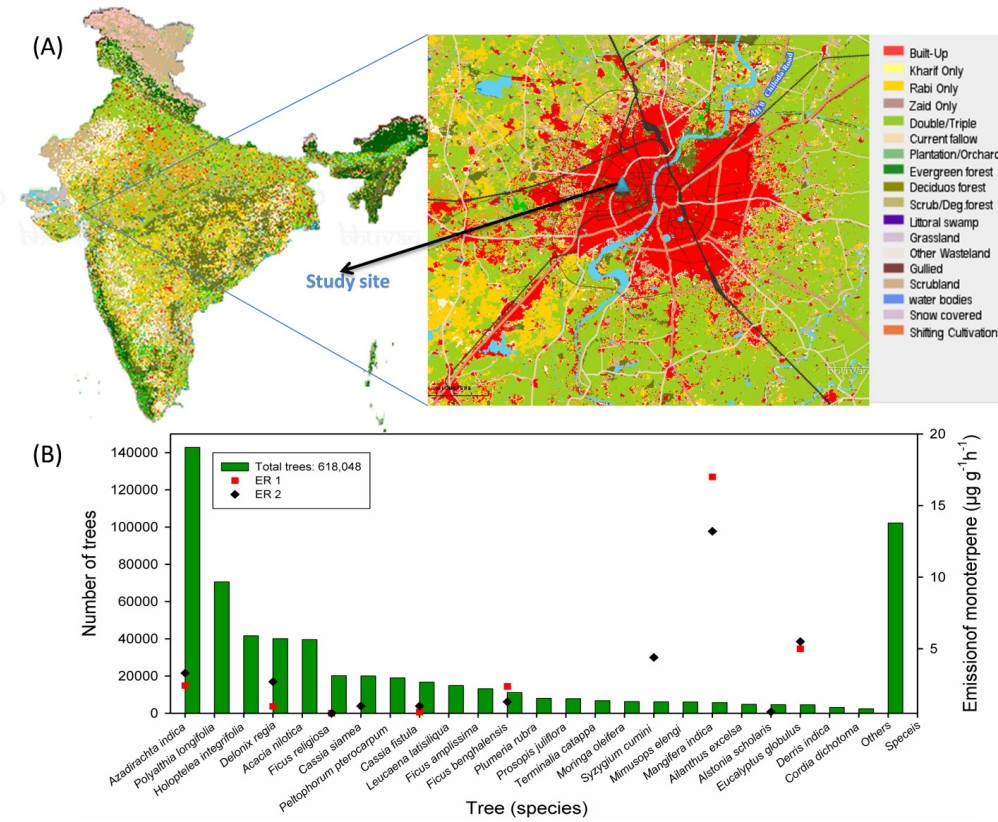


**Figure 1:** Top panel (A) shows the map of India and land use pattern of Ahmedabad city and surrounding areas (prepared using Bhuvan, the ISRO's Earth visualization portal (https://bhuvan.nrsc.gov.in/) and bottom panel (B) shows the number of major trees in Ahmedabad city and emission rates of monoterpenes for different plants taken from Singh et al., (2011) as ER 1

and Malik et al. (2018) as ER 2.



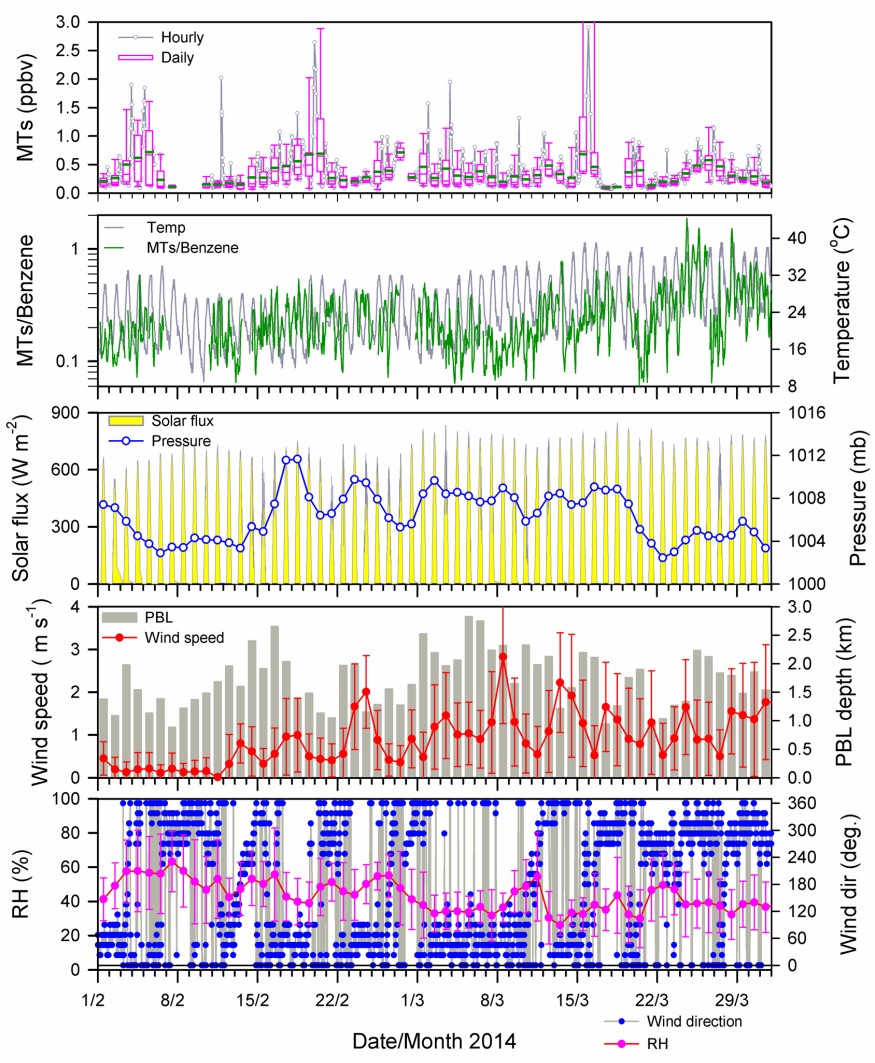


**Figure 2:** Time series of hourly and daily (box-whisker) monoterpenes (MTs) mixing ratio, MTs/benzene ratio, meteorological parameters and planetary boundary layer (PBL) depth at Ahmedabad from 01 February to 31 March 2014.





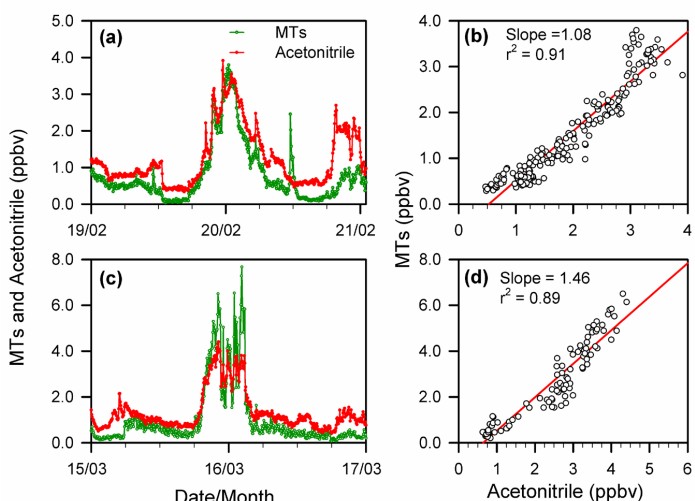

**Figure 3:** The time series of MTs and acetonitrile and their correlation plots during two major episodes of biomass burning (a, b) for 19-21 February and (c, d) for 15-16 March, *Holi* festival.






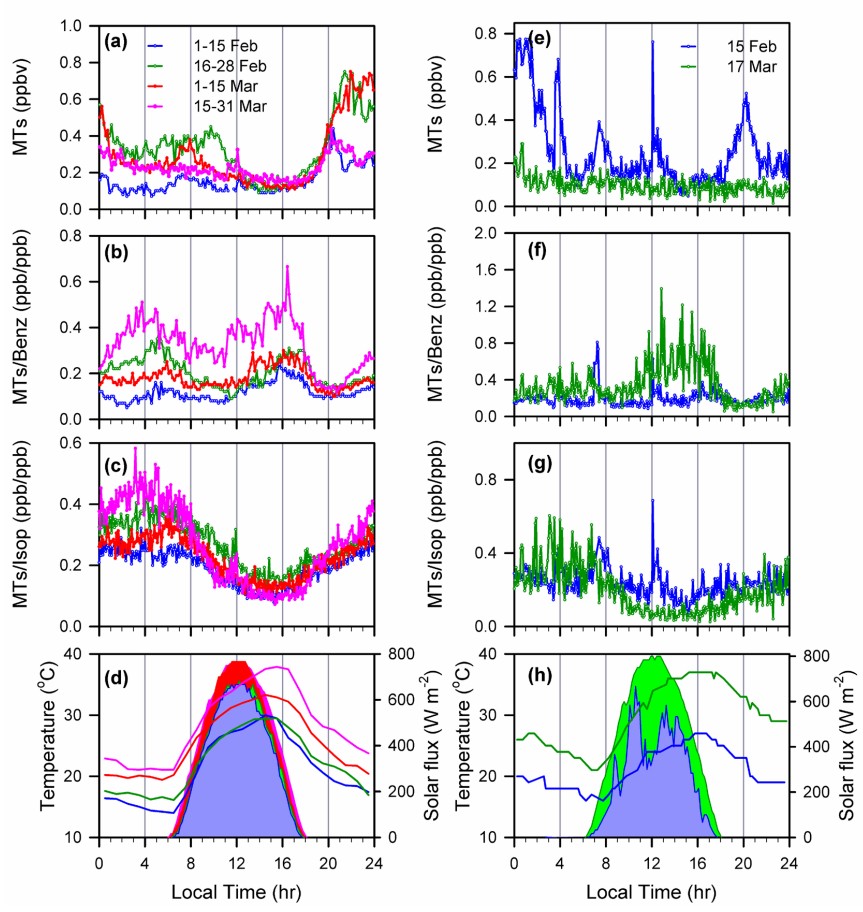


**Figure 4:** Average diurnal profiles of monoterpenes (MTs) mixing ratio, MTs/benzene ratio, MTs/isoprene ratio, temperature and solar flux during (a-d) four different periods of 1-15 February, 16-28 February, 1-15 March and 16-31 March 2014 and (e-h) for a cloudy day (15 February) and clear-sky day (17 March).





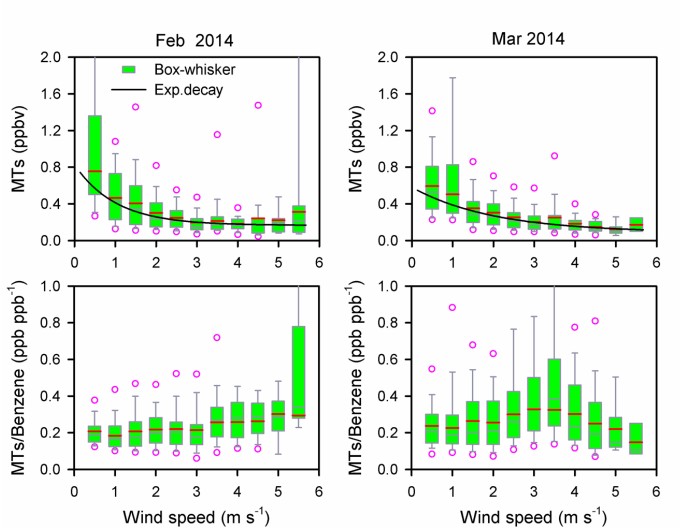


**Figure 5:** Variations of monoterpenes mixing ratio and monoterpenes/benzene ratio with wind speed during the months of February (left panel) and March 2014 (right panel).







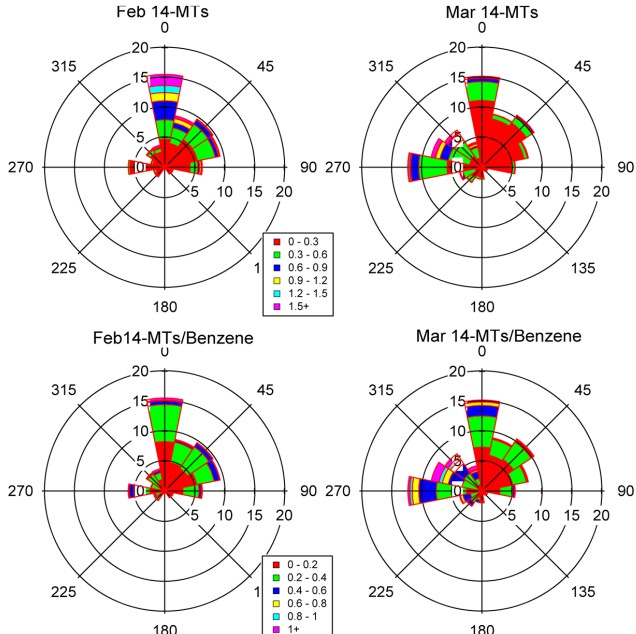

**Figure 6:** Polar frequency (%) plots of monoterpenes mixing ratio and monoterpenes/benzene ratio during the months of February 2014 (left panel) and March 2014 (right panel).



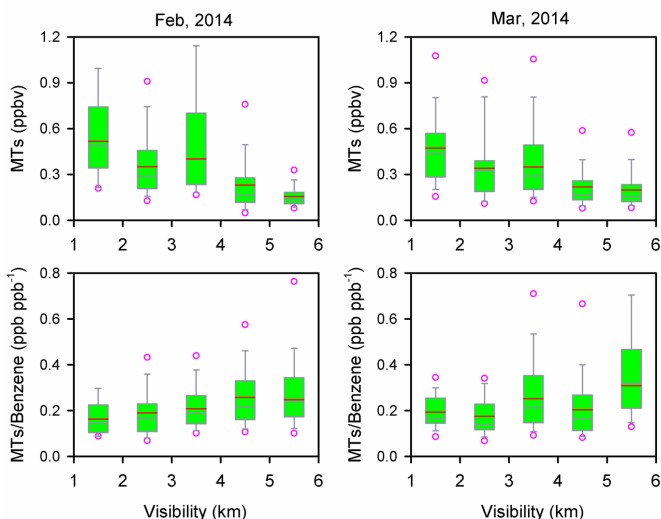

**Figure 7:** Variation of monoterpenes mixing ratio and monoterpenes (MTs)/benzene ratio with atmospheric visibility in February (left panel) and March 2014 (right panel).





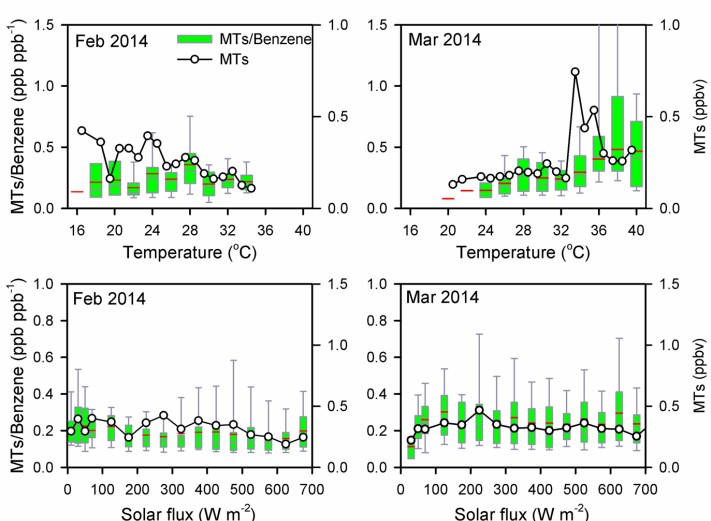

**Figure 8:** Dependencies of monoterpenes mixing ratio and monoterpenes/benzene ratio with ambient temperature and solar flux during February 2014 (left panel) and March 2014 (right panel).







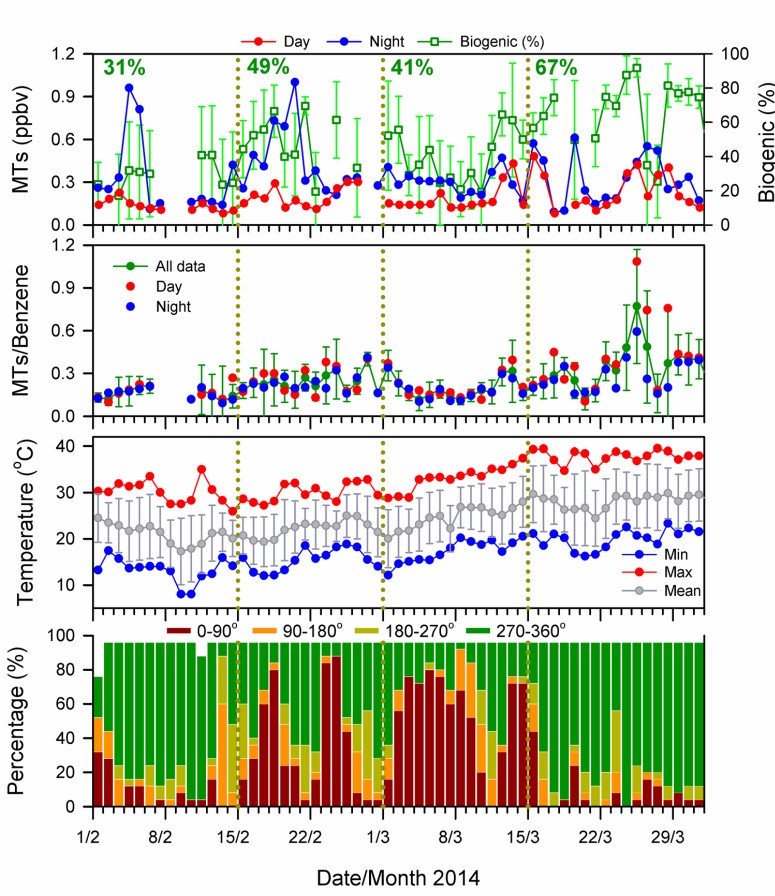

**Figure 9:** Time series of monoterpenes mixing ratio, monoterpenes/benzene ratio and estimated
biogenic contribution along with daily air temperatures and percentage occurrence of winds from
different sectors.





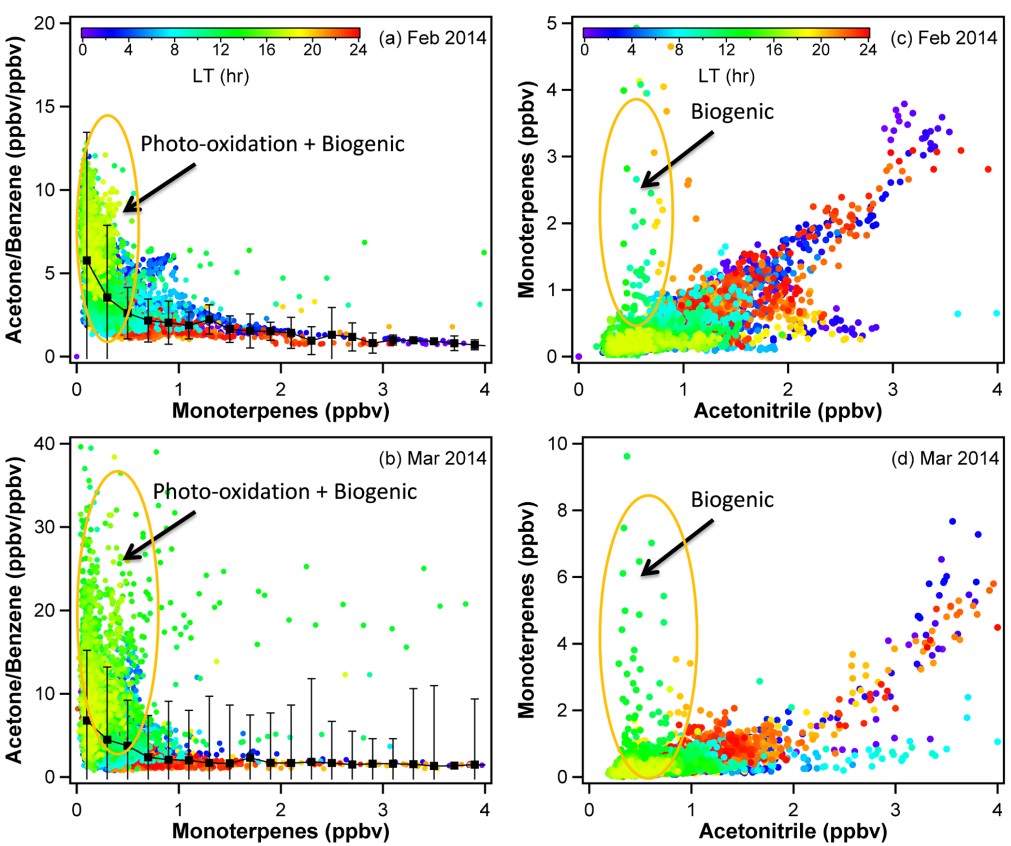

**Figure 10:** Relations between monoterpenes mixing ratio and acetone/benzene ratio during (a)
February and (b) March 2004 while (c) and (d) are scatter plots between mixing ratios of
monoterpenes and acetonitrile.