# Peer review of "Enhancement of biogenic emissions of VOCs in the semi-arid region of India during winter to summer transition period: Role of meteorological conditions"

_Atmospheric Chemistry and Physics, 2019_

## Referee Comment (RC1) · Anonymous Referee #1 · 19 Jun 2019

General Comments

This manuscript uses PTR-TOF-MS to measure monoterpenes for western India. The authors found high mixing ratios of monoterpenes from evening until midnight. The strong temperature is responsible for increasing the monoterpenes/benzene about 2 times. Additional, the increasing about 50% of local biogenic sources comes from regional transport from SE Asia.

While this manuscript provides information about biogenic VOCs emissions from local

[Figure]

India, I suggest the authors point the major innovations besides the study region in the manuscript to emphasize its significance of this work.

Specific Comments

Section 3.2, fig3 shows time series of monoterpenes from biomass burning. But due to February and March are strongly biomass burning emission time period in SE Asia, so it has a value to clarify how much is responsible for monoterpenes in local India.

Section 4.1, analysis of the effect of wind parameters on monoterpene using exponential decay functions and wind rose. The result will be stronger if the author could show the wind map that indicates the impact of Asian monsoon that brings the monoterpene from SE Asia.

---

## Referee Comment (RC2) · Anonymous Referee #2 · 25 Jun 2019

General Comments:

The manuscript entitled "Enhancement of biogenic emissions of VOCs in the semi-arid region of India during winter to summer transition period: Role of meteorological conditions" by Nidhi Tripathi and Lokesh Kumar Sahu reports high monoterpene mixing ratios using PTR-TOF-MS in Ahmedabad, India. The authors suggest that the monoterpenes mixing ratios show increasing trend from evening to midnight due to high temperature. They have also tried to establish the biogenic contribution to monoterpenes using monoterpenes/benzene emission ratios and from regional trans-

port. Although the manuscript provides information about biogenic and anthropogenic sources of monoterpenes from a part of the world where VOC data are scarce, it lacks scientific significance and needs to be presented in a more convincing way to the readers. The manuscript is not logically written and not well organized. It was very hard to understand what the main points are in the result and discussion section. Therefore, I recommend that the manuscript needs major revision and cannot be accepted for publication in ACP in the current format.

Specific Comments:

Title: The title "Enhancement of biogenic emissions of VOCs. . ." doesn't reflect the content of the manuscript in my opinion. In this manuscript, the authors tried to investigate the emission sources (that includes both biogenic and anthropogenic sources such as biomass burning and during Holi festival) of monoterpenes in the semi-arid region in India. As isoprene is the primary biogenic VOC and authors did not discuss the biogenic emission sources of isoprene in detail, I think the title does not justify the content of the manuscript and it needs to be revised.

Abstract: P1, L10-11: The authors themselves mentioned here that "This study is based on the measurements of monoterpenes" which was exactly what I pointed out in the Title of the manuscript.

P1, L12: The authors should mention the months here when they mention "winter-to-summer transition"

P1, L13-16: These sentences are kind of confusing to the readers. The authors first mentioned that "monoterpenes showed strong diurnal variation with elevated values from evening till midnight and lowest in the afternoon." Nighttime elevated monoterpenes mixing ratios cannot be attributed to biogenic emissions entirely as there can be contributions from other anthropogenic sources. This nighttime emission sources of monoterpenes need to be investigated in detail.

The next sentence they mentioned "The daily data does not show clear trends with monthly means of ∼0.35 ppbv during each month". This contradicts with the previous sentence where it is mentioned that "monoterpenes showed strong diurnal variation..." In addition, if we look at Table 1, the monthly means for February and March is different. Therefore, "monthly means of ∼0.35 ppbv during each month" doesn't make any sense.

Section 1: The authors should provide only the information relevant to the manuscript. If the authors are trying to focus only to address biogenic emissions, they should avoid providing information about anthropogenic emissions such as in P2, L40: "Major sources of anthropogenic VOCs (AVOCs) include combustion of fossil fuel, biomass burning, use of solvents, industrial production, refineries, etc.(Sahu, 2012)." and elsewhere.

The authors should also rearrange the information they want to provide in the Introduction section. Currently, it reads like several information gathered from previous works and the authors put that in the introduction section but most of the places there is no connection between the previous sentence and the next one which confuses the readers and is hard to follow.

Additionally, the authors should cite some previous work performed in urban areas in South Asia using PTR-TOF-MS e.g. Sarkar et al. (2016) reported a valuable dataset from urban Kathmandu Valley using PTR-TOF-MS.

Sarkar, C., Sinha, V., Kumar, V., Rupakheti, M., Panday, A., Mahata, K. S., Rupakheti, D., Kathayat, B., and Lawrence, M. G.: Overview of VOC emissions and chemistry from PTR-TOF-MS measurements during the SusKat-ABC campaign: high acetaldehyde, isoprene and isocyanic acid in wintertime air of the Kathmandu Valley, Atmos. Chem. Phys., 16, 3979-4003, https://doi.org/10.5194/acp-16-3979-2016, 2016.

Section 2: Authors should provide detailed information regarding the PTR-TOF-MS calibrations i.e. how many calibrations were performed during the measurement period, details of the standard and dilution flows of GCU, sensitivity plots, how the sensitivities

varied during the measurement period and effect of RH on MT sensitivity.

How was the transmission curve of the PTR-TOF looks like?

How was the estimation of monoterpenes mixing ratios using at m/z 137.131 performed since monoterpenes fragmentation also gives signal at 81.070?

How often was the zero-air test performed in a day and during the measurement period?

All this detailed information should be there (within the manuscript or as a supplementary information) to establish that the data presented in the manuscript is reliable.

Section 3.1: Is the measurement site $\alpha$-pinene dominant? Monoterpenes fragmentation pattern depends on instrumental condition as well as different monoterpene species as shown in Tani et al., 2004. Thus, if this site is characterized as an $\alpha$-pinene dominant area, the uncertainty in estimating MT concentration can be minimized. Otherwise, m137.131 reported in this study will imply a big uncertainty. Therefore, if this is the former case, please provide appropriate references or data.

Section 3.2: P8, L198: 16 March

P8, L202-205: What about the MT/acetonitrile ratios during nighttime for rest of the measurement period? The authors are emphasizing here on the emission of MT from biomass burning during the evening till next early morning during the Holi festival. However, in many places of the manuscript, the authors ignored this fact of biomass burning contribution to MT mixing ratios and emphasizing only on emissions from storage pool from plants due to high temperature during nighttime. The activity of biomass and wood burning is a common practice in India at evening and nighttime during wintertime.

Section 3.3: P10, L246-250: The statement "Therefore, in the absence of sunlight, higher MTs/isoprene ratios indicate higher nighttime biogenic emissions of monoterpenes than those of isoprene." is absurd as biogenic emission of isoprene do not occur at night.

[Figure]

The authors should explain how they estimated isoprene mixing ratios as in urban areas isoprene could be overestimated due to isomers from other sources. For instance, in smoke the "isoprene peak" is 20% pentadiene + cyclopentene.

What about the contribution of 232-MBO to isoprene mixing ratios?

It is well known that isoprene emissions can occur from traffic and BB and that can rationalize nighttime isoprene and contribute to daytime isoprene. Isoprene can have $\sim$ 20% interferences though from other compounds even at high mass resolution (Yokelson et al. 2013, Sarkar et al, 2016) in fresh smoke. Can the author estimate the fraction of observed isoprene from vegetation and combustion?

P11, L263: MT/isoprene ratios during early morning clearly indicates that there is anthropogenic contribution to the MT mixing ratios.

Section 4.1: P12, L290-293: It was unclear to me what authors are trying to say here since both the sentences read contradictory to me.

P13, L332-333: "Mixing ratios of monoterpenes were high in the lower visibility conditions (< 4 km) but showed lower values under high visibility conditions (> 4 km)" Shouldn't it be the opposite since reduction of visibility will cause after the photooxidation of MT?

Figure 4. a) and Figure 5: The diurnal profile and variations of nighttime MT mixing ratios seems highest from 16-28 February as compared to the other periods which contradicts the conclusions drawn by the authors that MT mixing ratios are higher during March than February.
* * *

---

## Referee Comment (RC3) · Anonymous Referee #3 · 28 Jun 2019

General comments:

The paper entitled "Enhancement of biogenic emissions of VOCs in the semi-arid region of India during winter to summer transition period: Role of meteorological conditions" by Tripathi and Sahu reports PTR-TOF-MS measurements of monoterpenes from a city in India during the period 1.02.2014 to 31.03.2014 and concluding that biogenic emissions increased in the transition from winter to summer. I was excited to see the title and new dataset but after going through the present manuscript and previous cited PTRTOFMS works by the same group, I realized a similar dataset (or

same dataset except for the monoterpene data shown here with similar sounding title "Contribution of biogenic and photochemical sources to ambient VOCs during winter to summer transition at a semi-arid urban site in India" has already been published in the journal Environmental Pollution in 2017. The authors cite this work in the present submission where they state they used benzene and isoprene as supporting data (Lines 143) but I could not find any discussion of novelty upon the previous dataset expect for reporting signals measured by the authors at m/z 137. What was more disconcerting about the submission is that the main methods and analyses presented in the work are seriously flawed (please see specific comments below for results and discussion section).

The present manuscript lacks a cohesive structure, makes tall claims not backed by hard evidence and has loose statements. It is riddled with claims that are at times even illogical. For example by simply having a rise in ambient temperatures and presence of some vegetation, one cannot attribute increase in monoterpenes to rising biogenic emissions in an atmospheric environment which has perhaps even stronger anthropogenic sources of monoterpenes (from varied types of biomass burning such as garbage fires and leaf litter burning to name a few). The so called quantitative methodology applied by the authors which assumes terpenes to be biogenic emissions and relies on inter VOC ratios to benzene, a molecule that has much longer chemical lifetimes relative to the terpenes and hence higher accumulation tendency is deeply flawed for application in such a complex emission environment. The authors highlight that the PTR-TOF-MS system enabled them to acquire highly mass resolved measurements. However the information and analyses they have presented concerning monoterpenes in the work nowhere makes use of this instrumental advantage and infact the information they show is even less well analysed than that acquired using a lower mass resolution PTR-MS. They do not use the high mass resolving power to unravel monoterpenes fragmentation to even speculate on the indentity of the monoterpenes and do not even discuss the major fragment at m/z 81, which most monoterpenes like alpha pinene yield in a PTR-MS system. This is poor use of the instrumentation. Disturbingly

the data quality control description also does not provide sufficient confidence that the measurements performed by the authors were done carefully and hence can be trusted, and are reliable.

The novelty of getting new data from a poorly sampled region on monoterpenes could have been the saving grace but even on this point concern about the quality of measurements and lack of novelty of the dataset in view of the previous published dataset puts a question mark on the utility of this work. The conclusion of increase in biogenic emissions and the title (highly misleading!) are not at all justified by the work presented in the manuscript. These points are elaborated using specific instances in the manuscript. Unfortunately considering the overall poor quality of the submission publication of the manuscript in ACP is not recommended.

Specific comments:

Introduction:

It is not well focused. Literature review of previous work is incomplete. For example in Line 89-90: The authors omit several important previous works (e.g. Sinha et al. 2014, Atmos Chem Phys) that have published isoprene data from India previously using PTR-MS including reporting the presence of strong biogenic and anthropogenic isoprene emitting sources, which highlighted that the city environments in South Asia are complex emission environments. These issues are therefore important to consider while using single molecular tracers in a quantitative manner as has been done by the authors.

Section 2: Measurement site Measurement site and PTR-TOF-MS instrumentation

Lines 100: It is clearly mentioned that car exhaust is a major source influencing the site, however subsequent analyses ignores this confounding influence on BVOC emissions as this source could explain most of the observed monoterpenes and isoprene.

Lines 110: Authors list some major tree species found in the area but do not describe

whether they are only monoterpene emitters or both isoprene and monoterpene emitters. If they are implicating such vegetation by using such leading remarks then why has isoprene only been reported superficially and just for MT/isoprene ratio calculations in this work?

Line 115: What about the role of agricultural emissions? Line 120: This information is useful but does not exclude the likelihood of anthropogenic sources in the city from combustion of varied biomass sources that are also upwind of the measurement site and closer to it mixing in additional terpene and VOC emissions.

Line 134: What good is high mass resolution of PTR-TOF-MS in this work when used to report only one ion, which is blindly ascribed to monoterpenes without any scrutiny? Also some monoterpenes can fragment and yield signals at m/z 79 (benzene? How have the authors accounted for such an effect if any? This is particularly critical as they use benzene signal as a purely anthropogenic tracer ! Line 137-139: Levels reported in the work are lower than 3 ppb so what good is knowing precision error at the such high values?

What about measurements to determine the instrumental background? How often were zeroes done? This is very important to know considering that lowest levels claimed to have been reported in the work were below 30 ppt. How were detection limits determined?

Line 141-143: Then why have the details and data for isoprene and benzene not been shown in same manner as the monoterpene data? Such fragmented approach to data usage and analysis is not desirable and encourages piecemeal approach to science.

Results and discussion:

Line 151: Weak winds were associated with higher mixing ratios...this could also be explained by more proximate anthropogenic sources like the road rather than transport of biogenic emission from the more distant forest. ...

Line 160-161: Municipal waste burning (see Stockwell et al. Atmos Chem Phys 2015) can also co-emit the terpenes and benzene. . .hence these cannot be used as exclusive tracers as the authors have done so . . ..Also owing to different chemical lifetimes (hours to minutes for monoterpenes and isoprene and several days for benzene) the following assertion by the authors is not tenable: "However, to some extent, the ratio of monoterpenes to benzene (an 160 anthropogenic tracer) can take account of variations due to change in local meteorology and PBL"

Lines 161-163 and Table 1: "Hourly monoterpenes/benzene ratio exhibits large periodic variation which tends to follow the diurnal cycle of temperature. Monoterpenes/benzene ratio showed slightly increasing trend with average values of 0.19±0.03 and 0.26±0.07 ppbv ppbv-1 during first and second halves of February, respectively."

The explanation linking higher temperature in March and increased MT/benzene ratios to increased biogenic emissions is deeply flawed. The ratio does not have to increase just because of numerator's value increasing..in fact it can also increase if denominator decreases and numerator stays constant! Benzene mixing ratios could very well decrease because the open biomass burning that occurs in winter for domestic heating by people without access to clean energy sources may have reduced in intensity during the transition from winter to summer due to warmer conditions. . .in fact this seems more likely based on the site description and city population than the attribution to biogenic sources. . . Why have the authors not reported and shown the one minute (they have highly time resolved data) benzene, isoprene, monoterpene, acetonitrile and MT data and its average mixing ratios on same axis for the full period? These would have been helpful to gauge what was really going on. From Figure 2 also looking at the available data it is clear that this is the more likely reason for increase in MT/benzene ratios.

As the basic premises and assumptions on which the further calculations and analyses have been presented (e.g. Figure 9 ) are unsound, the authors' conclusion and findings are deeply flawed.

[Figure]

Note from Copernicus Publications: The last sentence of this comment has been removed on 13 December 2019 on request by the ACP executive editors.

---

## Author Comment (AC1) · 9 Aug 2019

Response to Referee Comments along with revised version (edit mode) is uploaded as supplement.

Please also note the supplement to this comment: https://www.atmos-chem-phys-discuss.net/acp-2019-335/acp-2019-335-AC1-supplement.zip

---

## Author Comment (AC2) · 9 Aug 2019

Response to Referee Comments along with revised version (edit mode) is uploaded as supplement.

Please also note the supplement to this comment: https://www.atmos-chem-phys-discuss.net/acp-2019-335/acp-2019-335-AC2-supplement.zip

---

## Author Comment (AC3) · 9 Aug 2019

**Response to Anonymous Referee #3**

**The point-to-point answers (bold-font) of the comments (Referee # 3) are given here. The revised manuscript also includes the revisions suggested by two other referees.**

Referee Comment:
General comments: The paper entitled "Enhancement of biogenic emissions of VOCs in the semi-arid region of India during winter to summer transition period: Role of meteorological conditions" by Tripathi and Sahu reports PTR-TOF-MS measurements of monoterpenes from a city in India during the period 1.02.2014 to 31.03.2014 and concluding that biogenic emissions increased in the transition from winter to summer. I was excited to see the title and new dataset but after going through the present manuscript and previous cited PTRTOFMS works by the same group, I realized a similar dataset (or same dataset except for the monoterpene data shown here with similar sounding title "Contribution of biogenic and photochemical sources to ambient VOCs during winter to summer transition at a semi-arid urban site in India" has already been published in the journal Environmental Pollution in 2017. The authors cite this work in the present submission where they state they used benzene and isoprene as supporting data (Lines 143) but I could not find any discussion of novelty upon the previous dataset expect for reporting signals measured by the authors at m/z 137. What was more disconcerting about the submission is that the main methods and analyses presented in the work are seriously flawed (please see specific comments below for results and discussion section).

**Authors Reply: The general comments about the present manuscript in reference to our previous paper (Sahu et al., 2017) are completely incorrect and a non-issue. In disagreement with the referee comments we provide following facts which can be easily verified:**

**(1) This manuscript is all about monoterpenes. Is a single bit of monoterpene data reported in Sahu et al. (2017)?  Our answer is NO.**

**(2) In the present manuscript: Do we find any discussion of "isoprene concentration" Our answer is NO as it is used only as a reference.**

**(3) In present manuscript (conclusion section): Except one sentence (Lines 463-465, original Manuscript): "The higher nighttime ratios of monoterpenes/isoprene indicate significant light-independent but temperature dependent emissions of monoterpenes." Do we find the word "isoprene"?  Our answer is NO.**

**(4) A specific comment by the reviewer "isoprene only been reported superficially and just for MT/isoprene" also contradicts the general remark in reference to Sahu et al., 2017.**

**Despite very limited use we declared the use of other VOCs (isoprene, benzene) data in our original submission (Line 143, original MS). So where is further or reanalysis of data presented in Sahu et al., 2017?**

In case, if confusion is arising due to the Title, we have changed it (as also suggested by another referee). Now, the Title is changed to "Emissions and atmospheric concentrations of monoterpenes in a semi-arid region of India: Role of winter to summer changes in meteorological conditions"

Technical comment "..dataset expect for reporting signals measured by the authors at m/z 137.." by reviewer is completely incorrect. We request, the respected Editor to verify if reviewer's statement is a true reflection of what is reported in our paper (Line 134-135). Comments treating high mass resolution PTR-ToF-MS 8000 data in similar line to those with PTR-QMS (which we do not have) is factually incorrect. Following is our explanation and proof:

First of all we have not mentioned anything like 'reporting signals measured at m/z 137' in our paper? Therefore, according to this comment, in PTR-TOF-MS 8000 there are "NO" differences in following:

(1) between "m/z 137" and what we have reported in this paper "measured at m/z 137.131 (C10H16-H+)." Our answer is "YES" there is a difference, as monoterpene (α-pinene) was measured at 'm/z 137.131' along with products attributed to it but not at 'm/z 137'. See following, an example snapshots for m/z 137.131 and its main product at m/z 81.07 obtained from our PTR-TOF-MS during the study period (One can see the dates on spectra files so to avoid further doubts, if spectra were measured during the study period or not?). More information is provided as supplementary material.

[Figure]

 (2) between "reporting signals" and our calibrated data mentioned as "A certified gas mixture containing 0.97 ppmv ± 5% of α-pinene (L5388,Ionicon Analytik GmbH Innsbruck) was used for the calibration and determination of sensitivity." Our answer is "YES" there is a difference between "signal" and "calibrated data". Signals at m/z 137.131 and product (81.07) were calibrated with a standard mixture containing only one monoterpene (α-pinene) among other VOCs. Therefore, in the calibration, the product

detected at m/z 81.07 was only due to *α*-pinene and does not represent contributions or interferences of other monoterpenes as they were not present in the standard (calibration) mixture. Hence, the quality of data presented in this manuscript is very high and reliable.

Therefore Referee's comment "reporting signals measured by the authors at m/z 137." is incorrect. Implying a false sense that the data is not calibrated and wrong masses were used. Similarly, several facts have been incorrectly considered (e.g., mass resolving power).

Referee Comment: The present manuscript lacks a cohesive structure, makes tall claims not backed by hard evidence and has loose statements. It is riddled with claims that are at times even illogical. For example by simply having a rise in ambient temperatures and presence of some vegetation, one cannot attribute increase in monoterpenes to rising biogenic emissions in an atmospheric environment which has perhaps even stronger anthropogenic sources of monoterpenes (from varied types of biomass burning such as garbage fires and leaf litter burning to name a few). The so called quantitative methodology applied by the authors which assumes terpenes to be biogenic emissions and relies on inter VOC ratios to benzene, a molecule that has much longer chemical lifetimes relative to the terpenes and hence higher accumulation tendency is deeply flawed for application in such a complex emission environment.

Authors Reply: The assumption by the referee that anthropogenic (biomass burning) emission is not considered in the present study is factually incorrect and hence most of the reiterated comments. Otherwise, what is the meaning of "In Eq. (3), $MT_A$ represents local contribution mainly from biomass/bio-fuel burning" in Line 386 (original MS)? And, how we get about 70% MTs from anthropogenic sources during the first half of Feb (see Figure 9, original MS)?

If vegetations exist in any region irrespective of environments (semi-arid /forest/urban/rural) they respond to rapid changes in controlling parameters (mainly met/weather) hence can lead to significant changes in biogenic emissions of VOCs including monoterpenes. Precisely, this is the objective of the present work in a region where several millions tree exist in semi-arid tropical environments. All the details about trees/vegetation are given in "2. Measurement site and PTR-TOF-MS instrumentation". Following points already given in the original manuscript do not support the comments by the reviewer in particular "presence of some vegetation".

As mentioned in the manuscript (Lines 108-109; original MS), about $6.18 \times 10^5$ trees were counted in the year 2011 with relatively higher coverage in the western and south parts than those in eastern and north parts. This number ($6.18 \times 10^5$) excludes trees in surrounding nearby regions: As mentioned (Line 116-117, original MS): the Thol wildlife sanctuary is located ~ 25 northwest (NW) of site. As mentioned in Lines 118-119, Gandhinagar, also known as India's tree capital which has a green cover of about 54%, is located ~23 km north of site. See the Government document

**(https://forests.gujarat.gov.in/writereaddata/images/pdf/Status-of-Tree-Cover-in-Urban-Areas-of-Gujarat.pdf**) and (Govindarajulu, 2014; Vyas 2014).

**Vyas, D. N.: Floristic and ecological studies of Thol Lake Wildlife Sanctuary North Gujarat, 2014.**

**Govindarajulu D ( 2014) Urban Green Space Planning for Climate Adaptation in Indian Cities. Urban Climate 10, 35– 41.**

**About garbage fires and leaf litter burning: The policies of local governments are very strict and biomass burning such as garbage fires and leaf litter burning are prohibited. Nonetheless, CH$_3$CN (biomass marker) data do not suggest significant impacts of local garbage/litter burning except some events including *Holi* festival. The scenario is very different than those in the Indo-Gangetic Plain (IGP) and several other parts of the country. Even in case, such fires exist these were taken account (In Eq. (3), MT$_A$ represents local contribution mainly from biomass/bio-fuel burning, see Line 386 in original MS), otherwise how we get about 70% MTs from anthropogenic sources during the first half of Feb (see Figure 9, original MS)?**

**About the method:**
**Basically the methodology, also known as a source-tracer-ratio (STR) method and is not something new but has been widely used (e.g., Legreid et al., 2007; Yuan et al., 2012). For example, Goldstein et al., (2000) and Chang et al.,(2014) have applied STR to separate the contributions of anthropogenic and biogenic contributions of VOCs.**
**As the question is raised about the validity of this method, we describe it again here (which can be found in many research papers including the present manuscript). The measured emission ratio (ΔX/ΔY) of a compound (X) with respect to a reference compound (Y) has been widely used to estimate the emissions of many trace gases including VOCs (e.g., Yuan et al., 2012; Borbon, et al. 2013). Where X and Y are two different compounds and have different atmospheric lifetimes except a few cases. If the measurement locations (remote ocean or FT) are far from the source region the measured 'ratio (X/Y)' have been used as emission ratio (ΔX/ΔY) by applying correction due to photochemical loss known as age-corrected emission ratio (e.g., de Gouw et al., 2001; Baker et al., 2011; Thorenz et al., 2017). And purpose is to estimate the contribution for a specific sector or a source region (emission inventory). The emission ratios (e.g. monoterpene/isoprene) have been also used for the inventory developments of different BVOCs. In the present study, the emission ratio of MTs/benzene were estimated using evening data which are not significantly altered due to oxidation processes (see Lines 394-395), therefore does not require age-correction. But the age-correction particularly during the daytime will lead to increase of MTs/benzene ratios. Therefore, we have explicitly mentioned "underestimates the enhancement caused by biogenic emissions" in absence of age-correction. Consequently, biogenic contribution to ambient MTs can be higher than the estimated but the relative change from Feb to March will not be impacted significantly.**
**This methodology using ratios wrt references species to assess the biogenic contribution has been reported in several peer reviewed journals (Wagner and Kuttler, 2014, Filella, I. and Penuelas, 2006). In the present study, the applicability of MT/benzene ratio has been also**

assessed or supported by the additional use of MTs/isoprene ratio (this is why we are using isoprene data reported in Sahu et al., (2017). The MTs/isoprene ratio has been used in many studies to understand the biogenic emissions of VOCs. Measurements monoterpene to isoprene ratio was used to study the biogenic emission fluxes in a South-East Asian tropical rainforest (Langford, 2010, Jones et al., 2011, Misztal et al., 2011). Monoterpene emissions were also estimated from the isoprene emissions (e.g., Park et al., 2014). Monoterpene/isoprene emission factors for trees, grass, and shrubs for reported using the LPJ-GUESS model (Rap et al., 2018). About the use of ratio, limitation and implications have been explicitly discussed in original manuscript (see Lines 229-236, original manuscript): Overall, we have explained point by point that the methodology is correct and used by many researchers. And the reviewer's comment that biogenic sources make little or no contributions to ambient MTs is incorrect.

References (A few are also included in revised version):

Baker, A. K., Schuck, T. J., Slemr, F., van Velthoven, P., Zahn,A., and Brenninkmeijer, C. A. M.: Characterization of non-methane hydrocarbons in Asian summer monsoon outflow ob-served by the CARIBIC aircraft, Atmos. Chem. Phys., 11, 503–518,doi:10.5194/acp-11-503-2011, 2011.

Borbon, A., et al. ( 2013), Emission ratios of anthropogenic volatile organic compounds in northern mid-latitude megacities: Observations versus emission inventories in Los Angeles and Paris, J. Geophys. Res. Atmos., 118, 2041– 2057, doi:10.1002/jgrd.50059.

Chang, C. C., Wang, J. L., Leung, S.-C. C., Chang, C. Y., Lee, P.-J., Chew, C., Liao, W.-N., and Ou-Yang, C.-F.: Seosonal char-acteristics of biogenic and anthropogenic isoprene in tropical-subtropical urban environments, Atmos. Environ., 99, 298–308,2014.

de Gouw, J. A., Warneke, C., Scheeren, H. A., van der Veen, C.,Bolder, M., Scheele, M. P., Williams, J., Wong, S., Lange, L.,Fischer, H., and Lelieveld, J.: Overview of the trace gas mea-surements on board the Citation aircraft during the intensive fieldphase of INDOEX, J. Geophys. Res., 106, 28 453–28 467, 2001.

Filella, I. and Penuelas, J.: Daily, weekly, and seasonal time courses of VOC concentrations in a semi-urban area near Barcelona, At-mos. Environ., 40, 7752–7769, 2006.

Goldstein, A. H. and Schade, G. W.: Quantifying biogenic and anthropogenic contributions to acetone mixing ratios in a rural environment, Atmos. Environ., 34, 4997–5006, 2000.

Jones, C. E., Hopkins, J. R., and Lewis, A. C.: In situ mea-surements of isoprene and monoterpenes within a south-east Asian tropical rainforest, Atmos. Chem. Phys., 11, 6971–6984,doi:10.5194/acp-11-6971-2011, 2011.

Legreid, G., Lööv, J. B., Staehelin, J., Hueglin, C., Hill, M., Buchmann, B., Prevot, A. S. and Reimann, S.: Oxygenated volatile organic compounds (OVOCs) at an urban background site in Zürich (Europe): Seasonal variation and source allocation, Atmospheric Environment, 41(38), 8409–8423, 2007.

Langford, B., Misztal, P. K., Nemitz, E., Davison, B., Helfter, C.,Pugh, T. A. M., MacKenzie, A. R., Lim, S. F., and Hewitt, C. N.:Fluxes and concentrations of volatile organic compounds from a South-East Asian tropical rainforest, Atmos. Chem. Phys., 10,8391–8412,doi:10.5194/acp-10-8391-2010, 2010.

Misztal, P. K., Nemitz, E., Langford, B., Di Marco, C. F., Phillips,G. J., Hewitt, C. N., MacKenzie, A. R., Owen, S. M., Fowler,D., Heal, M. R., and Cape, J. N.: Direct ecosystem fluxes ofvolatile organic compounds from oil palms in South-East Asia,Atmos. Chem. Phys., 11, 8995–9017, doi:10.5194/acp-11-8995-2011, 2011.

Park, M. E., Song, C. H., Park, R. S., Lee, J., Kim, J., Lee, S.,Woo, J.-H., Carmichael, G. R., Eck, T. F., Holben, B. N., Lee,S.-S., Song, C. K., and Hong, Y. D.: New approach to monitortransboundary particulate pollution over Northeast Asia, Atmos.Chem. Phys., 14, 659–674, doi:10.5194/acp-14-659-2014, 2014.

Rap, A., Scott, C. E., Reddington, C. L., Mercado, L., Ellis, R. J.,Garraway, S., Evans, M. J., Beerling, D. J., MacKenzie, A. R.,Hewitt, C. N., and Spracklen, D. V.: Enhanced global primaryproduction by biogenic aerosol via diffuse radiation fertilization, Nat. Geosci., 11, 640–644, https://doi.org/10.1038/s41561-018-0208-3, 2018.

Thorenz, U. R.; Baker, A. K.; Leedham Elvidge, E. C.; Sauvage, C.; Riede, H.; Velthoven, P. F. J. van; Hermann, M.; Weigelt, A.; Oram, D. E.; Brenninkmeijer, C. A. M.; Zahn, A.; Williams, J., Investigating African trace gas sources, vertical transport, and oxidation using IAGOS-CARIBIC measurements between Germany and South Africa between 2009 and 2011, 2017. Atmospheric environment, 158, 11-26. doi:10.1016/j.atmosenv.2017.03.021.

Wagner, P. and Kuttler, W.: Biogenic and anthropogenic isoprene in the near-surface urban atmosphere—A case study in Essen, Germany, Science of the Total Environment, 475, 104–115, https://doi.org/10.1016/j.scitotenv.2013.12.026, 2014.

Yuan, B., et al. (2012), Volatile organic compounds (VOCs) in urban air: How chemistry affects the interpretation of positive matrix factorization (PMF) analysis, J. Geophys. Res., 117, D24302, doi:10.1029/2012JD018236.

Referee Comment: The authors highlight that the PTR-TOF-MS system enabled them to acquire highly mass resolved measurements. However the information and analyses they have presented concerning monoterpenes in the work nowhere makes use of this instrumental advantage and infact the information they show is even less well analysed than that acquired using a lower mass

resolution PTR-MS. They do not use the high mass resolving power to unravel monoterpenes fragmentation to even speculate on the indentity of the monoterpenes and do not even discuss the major fragment at m/z 81, which most monoterpenes like alpha pinene yield in a PTR-MS system. This is poor use of the instrumentation. Disturbingly the data quality control description also does not provide sufficient confidence that the measurements performed by the authors were done carefully and hence can be trusted, and are reliable.

**Authors Reply: This is an incorrect assumption. It is a very standard practice to account for all products ions and this has been done by taking account the fragmentation at m/z 81.07 (so that sensitivity is estimated for m/z at 137.131 which takes account the response of fragments). See a following snapshot of relevant part of a mass spectrum obtained during the study period. The experts using PTR-TOF-MS will disagree with the reviewer's assumption and comments about mass resolving power particularly 'lower mass resolution PTR-MS' and also on monoterpenes fragmentation at m/z 81.07. This has been taken account in MTs data (see Supplementary information). Instead of presumption, it could have been better, if the referee had asked for further clarification wherever required.**

[Figure]

Referee Comment: The novelty of getting new data from a poorly sampled region on monoterpenes could have been the saving grace but even on this point concern about the quality of measurements and lack of novelty of the dataset in view of the previous published dataset puts a question mark on the utility of this work. The conclusion of increase in biogenic emissions and the title (highly misleading!) are not at all justified by the work presented in the manuscript. These points are elaborated using specific instances in the manuscript. Unfortunately considering the overall poor quality of the submission publication of the manuscript in ACP is not recommended.

**Authors Reply: In the original manuscript we have provided all basic information required to judge the quality of data. Surely, if one doubts that given information is not sufficient**

then we have to provide additional evidences. In any case, now we have also provided supplementary information about the experimental setup (calibration) and data. And revised section 2 (Lines 126-148 in Revised MS).

About the novelty: The works about the changes in monoterpene (an important BVOC) during winter to summer transition period have not been reported for other regions of India (to the best of our knowledge). If available but missed, we are sure to have received the references from this referee which we did not. One of the referees comment "..monoterpenes from a part of the world where VOC data are scarce.." clearly endorses the novelty of the manuscript. So we are not sure, on what basis the novelty of this study has been questioned?

Specific comments:

Introduction:

Referee Comment: It is not well focused. Literature review of previous work is incomplete. For example in Line 89-90: The authors omit several important previous works (e.g. Sinha et al. 2014, Atmos Chem Phys) that have published isoprene data from India previously using PTR-MS including reporting the presence of strong biogenic and anthropogenic isoprene emitting sources, which highlighted that the city environments in South Asia are complex emission environments. These issues are therefore important to consider while using single molecular tracers in a quantitative manner as has been done by the authors.

**Authors Reply: As mentioned that the main focus of this manuscript is monoterpene ($\alpha$-pinene) but not isoprene. Nevertheless, referring a particular research paper is very subjective, but a sense or feeling that this research group ignores certain works is not correct. The authors are not biased towards a particular research group, the reviewer may find or already aware of the fact that this suggested paper (Sinha et al. 2014, Atmos Chem Phys) has been cited in following research papers from our group.**

**Sahu, L. K. and Saxena, P.: High time and mass resolved PTR-TOF-MS measurements of VOCs at an urban site of India during winter: Role of anthropogenic, biomass burning, bio-genic and photochemical sources, Atmos. Res., 164–165, 84–94,https://doi.org/10.1016/j.atmosres.2015.04.021, 2015.**

**Ravi Yadav, L K Sahu, Nidhi Tripathi, D Pal, G Beig, SNA Jaaffrey, (2019), Investigation of emission characteristics of NMVOCs over urban site of western India, Environmental Pollution, 252, 245-255.**

**Chutia, L., Ojha, N., Girach, I.A., Sahu, L.K., Alvarado, L.M.A., Burrows, J.P., Pathak, B., Bhuyan, P.K. (2019), Distribution of volatile organic compounds over Indian subcontinent during winter: WRF-chem simulation versus observations, 252, 256-269.**

**However, found to be relevant as also suggested by the reviewer we have cited other papers including following one.**

**Sarkar, C., Sinha, V., Kumar, V., Rupakheti, M., Panday, A., Mahata, K. S., Rupakheti, D., Kathayat, B., and Lawrence, M. G.: Overview of VOC emissionsand chemistry from PTR-TOF-MS measurements during the SusKat-ABC campaign: high acetaldehyde, isoprene and isocyanic acid in wintertime air of the Kathmandu Valley,Atmos. Chem. Phys., 16, 3979-4003, https://doi.org/10.5194/acp-16-3979-2016, 2016.**

Section 2: Measurement site Measurement site and PTR-TOF-MS instrumentation

Referee Comment: Lines 100: It is clearly mentioned that car exhaust is a major source influencing the site, however subsequent analyses ignores this confounding influence on BVOC emissions as this source could explain most of the observed monoterpenes and isoprene.

**Authors Reply: Significant emissions of MTs and isoprene from both biogenic and anthropogenic sources were reported in several urban regions of the world (some references are given below). In the present manuscript we have never denied the role of anthropogenic emissions and this is clearly mentioned. In Figure 9 (original MS), biogenic contribution is only about ~31% in the first half of Feb, so definitely anthropogenic emissions have major contributions. But the objective of the paper is to investigate that how the contribution of biogenic emissions changed due to winter-summer transition in response to large change in met parameters. And like other studies (e.g., for Essen, Germany by Wagner and Kuttler, (2014), we have used source-ratio-tracer method to separate the contributions from distinct emission sources. Following is a sentence from above referred (Wagner and Kuttler, 2014) study "The isoprene concentration and the isoprene/benzene ratio decreased during September and the first half of October (Fig. 6), which can be explained by the seasonal drop in air temperatures and light intensity and the senescence of leaves."**

**Following Fig is a comparison of daytime enhancements of isoprene/benzene ratio measured for clear-sky and cloudy days at present site in winter (January 2014) when traffic exhaust accumulations were higher than the present study period.**

[Figure]

**Now questions are:**

**(1) Why isoprene/benzene ratio follows the cycle of ambient temp but not the traffic pattern?**
**(2) Why there should be lesser daytime enhancements on a cloudy day compared to a clear sky day?**
**Answer: Biogenic emissions of VOCs are present in the tropical urban region even in peak winter.**

**References:**

**Li, N., He, Q., Greenberg, J., Guenther, A., Li, J., Cao, J., Wang, J., Liao, H., Wang, Q., and Zhang, Q.: Impacts of biogenic and anthropogenic emissions on summertime ozone formation in the Guanzhong Basin, China, Atmos. Chem. Phys., 18, 7489-7507, https://doi.org/10.5194/acp-18-7489-2018, 2018.**

**Khan, M.A.H.; Schlich, B.L.; Jenkin, M.E.; Shallcross, B.M.A.; Moseley, M.; Waler, C.; Morris, W.C.; Derwent, R.G.; Percival, C.J.; Shallcross, D.E. A two-decade anthropogenic and biogenic isoprene emissions study in a London urban background and a London urban traffic site. Atmosphere 2018, 9, 387.**

**Owen, S. M., MacKenzie, A. R., Stewart, H., Donovan, R., and Hewitt, C. N. ( 2003), Biogenic volatile organic compound (VOC) emission estimates from an urban tree canopy, Ecol. Appl., 13, 927– 938.**

**Seco, R., Peñuelas, J., Filella, I., Llusia, J., Schallhart, S., Metzger, A., Müller, M., and Hansel, A.: Volatile organic compounds in the western Mediterranean basin: urban and rural winter measurements during the DAURE campaign, Atmos. Chem. Phys., 13, 4291-4306, https://doi.org/10.5194/acp-13-4291-2013, 2013.**

**von Schneidemesser, E., Monks, P. S., Gros, V., Gauduin, J., and Sanchez, O. ( 2011), How important is biogenic isoprene in an urban environment? A study in London and Paris, Geophys. Res. Lett., 38, L19804, doi:10.1029/2011GL048647.**

Referee Comment: Lines 110: Authors list some major tree species found in the area but do not describe whether they are only monoterpene emitters or both isoprene and monoterpene emitters. If they are implicating such vegetation by using such leading remarks then why has isoprene only been reported superficially and just for MT/isoprene ratio calculations in this work?
**Author's Reply: Although the objective of this paper is not to characterize the emissions from a particular type of plant or vegetation but this is an important point by reviewer. As**

shown in Figure 1, tree species like *Mangifera indica, Eucalyptus globules, Ficus benghalensis, Syzygium, etc.* are major *α*-pinene emitters for which emissions of isoprene were reported (e.g., Varshney and Singh, 2003; Padhy and Varshney, 2005 and references therein). So the biogenic emissions of *α*-pinene and isoprene co-exist, we have revised for the same as here: Please see following in Revised MS, Lines 104-108:

"In India, emissions of isoprene from common plant species have been examined to some extent (Singh et al., 2011; Varshney and Singh, 2003). As shown in Figure 1, tree species such as *Mangifera indica*, *Eucalyptus globulus*, *Ficus benghalensis*, *Syzygium*, etc. are significant *α*-pinene emitters for which emissions of isoprene were reported (e.g., Padhy and Varshney, 2005; Varshney and Singh, 2003 and references therein)."

Referee Comment: Line 115: What about the role of agricultural emissions?

**Authors Reply: In this manuscript, the objective is not to quantify or identify emission contribution from a particular type of tree or agricultural species. But the natural emissions from any of vegetations (tree, grass, agricultural plant, etc.) are regarded as biogenic. The purpose of providing details of vegetations in section 2 is to describe the study site and surrounding regions.**

Referee Comment: Line 120: This information is useful but does not exclude the likelihood of anthropogenic sources in the city from combustion of varied biomass sources that are also upwind of the measurement site and closer to it mixing in additional terpene and VOC emissions.

**Authors Reply: We do not understand the importance of this comment in a view that the role of anthropogenic emissions are considered and discussed in details (both in original and revised manuscripts). In this paper we have not denied the role of anthropogenic sources and there is a dedicated section (3.2 Impact of biomass burning) including Figure 3. We suggest, see Figure 10 (of original MS) and only 31% MTs is biogenic. However, anthropogenic emissions were taken care to estimate the change in contributions of biogenic emissions due to rapid and large changes in met and weather conditions associated with the winter-to-summer transition as explained below. In any case, see following self-explanatory Figure (based on Table 1) showing small changes in reference compounds (benzene and acetonitrile) but significant changes in MTs and MTs/Benzene due to winter-summer transition in biogenic emissions.**

[Figure]

Referee Comment: Line 134: What good is high mass resolution of PTR-TOF-MS in this work when used to report only one ion, which is blindly ascribed to monoterpenes without any scrutiny? Also some monoterpenes can fragment and yield signals at m/z 79 (benzene? How have the authors accounted for such an effect if any? This is particularly critical as they use benzene signal as a purely anthropogenic tracer!

**Author's Reply: Again incorrect presumption. In this study benzene was measured at m/z 79.0548 but not at m/z 79 (Sahu and Saxena, 2015). Kari et al., (2018) have provided product ion distributions of twelve different MTs at 130 Td similar to 128-130 Td used in the present study. In any case, Kari et al., do not report any products at m/z 79 (or even at m/z 79.0548) coming from any of 12 MTs (including α-pinene). For α-pinene, two masses at m/z at 137.131 and m/z at 81.07 account for about 96%. Therefore, benzene data (calibrated with known standard) used in the present study does not carry the interferences of MTs. In any case, more details are provided in supplementary information, if there is confusion about MTs data.**

**Kari, E., Miettinen, P., Yli-Pirilä, P., Virtanen, A., and Faiol, C. L. (2018). PTR-ToF-MS product ion distributions and humidity-dependence of biogenic volatile organic compounds. Int. J. Mass Spectrom. 430, 87–97. doi: 10.1016/j.ijms.2018.05.003.**

**Sahu, L. K., and P. Saxena (2015), High time and mass resolved PTR-TOF-MS measurements of VOCs at an urban site of India during winter: role of anthropogenic, biomass burning, biogenic and photochemical sources, Atmospheric Research, 164-165, 84-94, http://dx.doi.org/10.1016/j.atmosres.2015.04.021.**

Referee Comment: Line 137-139: Levels reported in the work are lower than 3 ppb so what good is knowing precision error at the such high values? What about measurements to determine the

instrumental background? How often were zeroes done? This is very important to know considering that lowest levels claimed to have been reported in the work were below 30 ppt. How were detection limits determined?

**Author's Reply: We believe, instead of average levels, it is the "range" which should be important to decide the reference for quantification of the precision/repeatability. Refer to Figure 3 (original manuscript), MTs values exceeding 6 ppbv were measured (even though such enhancements were limited only to a few events), therefore nothing wrong reporting precision at 3 ppbv. We can provide data for other levels but it's not clear what the reviewer wants? It is important to ensure the high stability of zero-air to determine the background hence to quantify the overall performance of PTR-TOF-MS including sensitivity as "zero air" is also used as a dilution gas in GCU. The stability of background level (zero) is also important for determination of the limit of detection (LOD). Therefore, instead of using ambient air (with variable VOC) as input in GCU, we connected high purity zero air generator (Parker HPZA-3500-220) in tandem with GCU. The instrumental background (zero) and calibration of PTR-TOF-MS were performed in about 2- and 10-day intervals, respectively. The limit of detection (LOD) determined as [2×(standard deviation of background/sensitivity)]. Above details are now provided as supplementary material.**

Referee Comment: Line 141-143: Then why have the details and data for isoprene and benzene not been shown in same manner as the monoterpene data? Such fragmented approach to data usage and analysis is not desirable and encourages piecemeal approach to science.

**Authors Reply: We have provided supplementary information for details of MTs measurement. Isoprene and benzene are used as supporting data for which details are reported in our previous publications (Sahu et al., 2015, Sahu et al., 2016). And same has been mentioned in the revised manuscript.**

**Sahu, L. K. and Saxena, P.: High time and mass resolved PTR-TOF-MS measurements of VOCs at an urban site of India during winter: Role of anthropogenic, biomass burning, bio-genic and photochemical sources, Atmos. Res., 164–165, 84–94,https://doi.org/10.1016/j.atmosres.2015.04.021, 2015.**

**Sahu, L., Yadav, R. and Pal, D.: Source identification of VOCs at an urban site of western India: Effect of marathon events and anthropogenic emissions, Journal of Geophysical Research: Atmospheres, 121(5), 2416–2433, doi:10.1002/2015JD024454, 2016b.**

Results and discussion:

Referee Comment: Line 151: Weak winds were associated with higher mixing ratios: : :this could also be explained by more proximate anthropogenic sources like the road rather than transport of biogenic emission from the more distant forest: : :.

**Authors Reply: We have explained the wind direction dependence of MTs presented in Figs 6 and 9 (original Manuscript). Again, we have not stated anywhere in the manuscript that anthropogenic (including biomass burning) emission is not contributing to the ambient MTs. In the first half of Feb, about 70% of ambient MTs originated from anthropogenic emissions (major source) [see Figure 9 of original Manuscript & Fig 10 in Revised Manuscript]. And wind direction dependence shown in Figure 6 (Figure 7 in Revised Manuscript) is not for just for biogenic MTs alone but it includes contributions from anthropogenic sources as well. So, there is nothing wrong in our statements about the wind direction dependence.**

Referee Comment: Line 160-161: Municipal waste burning (see Stockwell et al. Atmos Chem Phys 2015) can also co-emit the terpenes and benzene: : :hence these cannot be used as exclusive tracers as the authors have done so : : :.Also owing to different chemical lifetimes (hours to minutes for monoterpenes and isoprene and several days for benzene) the following assertion by the authors is not tenable: "However, to some extent, the ratio of monoterpenes to benzene (an 160 anthropogenic tracer) can take account of variations due to change in local meteorology and PBL"

**Authors Reply: First of all there is nothing like Municipal waste burning near the study site. Municipal waste is dumped (not burnt) at *Pirana Landfill Site* located about 10-12 km southeast (SE) of the study site. In any case of accidental fires, the Pirana site is not in the upwind (Proof: See wind plots Fig 7 (Revised MS), the SE winds were never measured during the study period). But yes, there were sporadic plumes of biomass burning mainly during the Holi festival (we have already described in details). We do not understand, what is the point here? We have not stated anywhere in the manuscript that anthropogenic (including biomass burning) is not contributing to ambient MTs. Therefore, how we are contradicting the suggested phrase "Municipal waste burning (see Stockwell et al. Atmos Chem Phys 2015) can also co-emit the terpenes and benzene" by the referee? In Figure 3 (Reviseds MS), the mixing ratio of benzene along with acetonitrile are plotted to trace the anthropogenic emissions of MTs. The strong correlations of MTs with benzene during sporadic biomass burning events further clarifies that benzene can be used as a tracer to mark or separate the anthropogenic contributions.**

[Figure]

**About the use of ratios, limitation and implications of results have been explicitly discussed in the original manuscript (see Lines 229-236, original MS) and revised manuscript as well. Overall, we have explained point by point that the methodology is correct and used by the researchers as reported in many peer reviewed papers.**

Referee Comment: Lines 161-163 and Table 1: "Hourly monoterpenes/benzene ratio exhibits large periodic variation which tends to follow the diurnal cycle of temperature. Monoterpenes/benzene ratio showed slightly increasing trend with average values of 0.190.03 and 0.260.07 ppbv ppbv-1 during first and second halves of February, respectively." The explanation linking higher temperature in March and increased MT/benzene ratios to increased biogenic emissions is deeply flawed. The ratio does not have to increase just because of numerator's value increasing..in fact it can also increase if denominator decreases and numerator stays constant! Benzene mixing ratios could very well decrease because the open biomass burning that occurs in winter for domestic heating by people without access to clean energy sources may have reduced in intensity during the transition from winter to summer due to warmer conditions: : :in fact this seems more likely based on the site description and city population than the attribution to biogenic sources: : : Why have the authors not reported and shown the one minute (they have highly time resolved data) benzene, isoprene, monoterpene, acetonitrile and MT data and its average mixing ratios on same axis for the full period? These would have been helpful to gauge what was really going on. From Figure 2 also looking at the available data it is clear that this is the more likely reason for increase in MT/benzene ratios.

**Reply: Referee's comment providing such simple numerical analysis (numerator/denominator) of the observables is incorrect as the comment has ignored the underlying emissions processes. There is a difference between "ratio" and "emission ratio". According to the referee the concept of "emission factor", "emission ratio" and their used in "emission model" which deal with numerator/denominator is wrong, is that the meaning of this comment? The relations (or dependencies) of both MTs and MTs/benzene ratio on CH$_3$CN using all data points measured during the months of February and March are**

**presented separately in Figure 4 (revised manuscript). This analysis clearly highlights the parts of data tagged as biogenic and anthropogenic.**

[Figure]

**In this paper we have not denied the role of anthropogenic sources and there is a dedicated section (3.2 Impact of biomass burning) including Figure 3. The anthropogenic emissions were taken account to estimate the change in contributions of biogenic emissions due to rapid and large changes in met and weather conditions associated with the winter-to-summer transition as explained. About reviewer comment: "open biomass burning that occurs in winter for domestic heating by people without access to clean energy sources", the authors disagree with this self contradictory statement as there is nothing like "open biomass burning" for "domestic heating". In any case, as shown in time series (Figure 2), warmer weather conditions (15-40$^o$C) prevailed over the study region. We are aware of the fact that people do not use "open biomass burning" for domestic heating in such warm conditions. If at all some warming is required, use of biomass burning in this one of the most developed (economically and technologically) regions of India is minimal.**

**Both weather and emission patterns are very different compared to those for Indo-Gangentic Plains (IGP) where activities of biomass burning are rampant in winter. We appreciate the works on isoprene reported in Sinha et al., 2014 over IGP. But comments on role of biomass burning for this study are not necessarily valid because of different emission patterns and also different weather conditions. Nevertheless, sporadic events of biomass burning were explicitly discussed in the manuscript. See a dedicated section (3.2 Impact of biomass burning) and Figure 3. Further, the relations of MTs with a biomass maker (acetonitrile, CH$_3$CN) support our argument in contrast with the reviewer's opinion (see Figure 4, revised manuscript). This clearly separates the contributions from other sources (at least biomass burning). We have used other VOC (isoprene, acetonitrile, etc.) just to support the analysis. We clarify, that we have not concluded anything on other species except monoterpene (see abstract and conclusion). About the use of MT/benzene ratios we have provided all details including limitations in response to previous comments. It would have been helpful if reviewer could have suggested some better ways of analyzing the data instead comments based on incorrect presumptions. If anything to infer about very short-term variations, well time resolved data are presented in Figures 3, 4 and 5 (revised MS). In reply to a previous comment, we have provided the use of source-tracer-**

**ratio method including limitations and implications. As mentioned in Line (405-406, original MS): "As reported in Sahu et al. (2017) that the mixing ratio of acetonitrile does not show any clear trend during the winter-summer period."  Now, using CH3CN and benzene as marker, what better can be the proof or fact that the impact of biomass burning was small & in any case, taken account for the estimated contributions biogenic MTs.**

Reviewer Comment: As the basic premises and assumptions on which the further calculations and analyses have been presented (e.g. Figure 9) are unsound, the authors' conclusion and findings are deeply flawed.

**Reply: We hope, our point by point replies on comments related to both the measurements and data analysis clarifies all the doubts raised by the reviewer.**

Note from Copernicus Publications: The last part of this reply has been removed on 13 December 2019 on request by the ACP executive editors.